



# The Far-Infrared Radiation Mobile Observation System for spectral characterisation of the atmospheric emission

Claudio Belotti[1], Flavio Barbara[2], Marco Barucci[1], Giovanni Bianchini[1], Francesco D'Amato[1], Samuele Del Bianco[2], Gianluca Di Natale[1], Marco Gai[2], Alessio Montori[1], Filippo Pratesi[1], Markus Rettinger[3], Christian Rolf[4], Ralf Sussmann[3], Thomas Trickl[3], Silvia Viciani[1], Hannes Vogelmann[3], and Luca Palchetti[1]

[1]National Institute of Optics, National Research Council, Florence, Italy
[2]Institute of Applied Physics "Nello Carrara", National Research Council, Florence, Italy
[3]Karlsruhe Institute of Technology (KIT), IMK-IFU, Garmisch-Partenkirchen, Germany
[4]Forschungszentrum Jülich, IEK-7, Jülich, Germany

**Correspondence:** Claudio Belotti (claudio.belotti@cnr.it)

**Abstract.** The Far-Infrared Radiation Mobile Observation System (FIRMOS) is a Fourier transform spectroradiometer developed to support the Far-infrared Outgoing Radiation Understanding and Monitoring (FORUM) satellite mission by validating measurement methods and instrument design concepts, both in the laboratory and in field campaigns. FIRMOS is capable of measuring the downwelling spectral radiance emitted by the atmosphere in the spectral band from 100 to 1000 $cm^{-1}$ (10–100 $\mu m$ in wavelength), with a maximum spectral resolution of 0.25 $cm^{-1}$. We describe the instrument design and its characterisation and discuss the geophysical products obtained by inverting the atmospheric spectral radiance measured during a campaign from the high-altitude location of Mount Zugspitze in Germany, beside the Extended-range Atmospheric Emitted Radiance Interferometer (E-AERI), which is permanently installed at the site. Following the selection of clear-sky scenes, using a specific algorithm, the water vapour and temperature profiles were retrieved from the FIRMOS spectra by applying the Kyoto protocol and Informed Management of the Adaptation (KLIMA) code. The profiles were found in very good agreement with those provided by radiosondes and by the Raman lidar operating from the Zugspitze Schneefernerhaus station. In addition, the retrieval products were validated by comparing the retrieved Integrated Water Vapour values with those obtained from the E-AERI spectra. Finally, we found that the trends for the temperature, and the water vapour profiles over time were in good agreement with those provided by ERA5 reanalysis.

## 1 Introduction

The far-infrared (FIR) portion of the Earth's emission spectrum is the subject of a growing research interest because of its important role played in the Earth's radiative balance. This spectral region covers the wavelengths longer than 15 μm (the





wavenumbers below $667\ \mathrm{cm}^{-1}$) and is strongly characterised by the pure rotational absorption band of water vapour and the $\nu_2$
carbon dioxide band. Several atmospheric and surface processes contribute to both the outgoing and the incoming radiation at
these wavelengths in a complex and entangled manner (Harries et al., 2008; Palchetti et al., 2020, see for a detailed discussion).
In this context, spectrally resolved radiometric observations are a valuable tool that can potentially quantify the role of each of
these contributions on the overall radiative balance.

To date, the FIR component of the outgoing longwave radiation has only been measured a few times during balloon cam-
paigns by REFIR-PAD (Palchetti et al., 2006) and FIRST (Mlynczak and Johnson, 2006), and by the airborne instrument
TAFTS (Cox et al., 2010). On the other hand, several ground-based experiments observed the FIR portion of the downwelling
longwave radiation (DLR): the Earth Cooling by Water Vapor Radiation (ECOWAR) experiment (Bhawar et al., 2008), and the
Radiative Heating in Underexplored Bands Campaigns (Turner and Mlawer, 2010; Turner et al., 2012, RHUBC-I and RHUBC-
II). Eventually, REFIR-PAD was installed in Antarctica at the Concordia station, where it has been in continuous operation
since 2011 (Bianchini et al., 2019).

FIR spectral measurements of DLR proved valuable for refining the knowledge of water vapour spectroscopy (Mlawer et al.,
2019) and testing the ability to model radiative transfer in the atmosphere (Mlynczak et al., 2016; Mast et al., 2017; Bellisario
et al., 2019; Mlawer et al., 2019). In addition, ground-based FIR observations were successfully exploited to infer cloud
properties (Maestri et al., 2014; Rizzi et al., 2016; Di Natale et al., 2017), to retrieve the thermal structure and composition
of the atmosphere (Rizzi et al., 2018; Bianchini et al., 2019), as well as to conduct radiative closure studies (Delamere et al.,
2010; Sussmann et al., 2016).

The Far-infrared Outgoing Radiation Understanding and Monitoring (Palchetti et al., 2020, FORUM) project has been
selected as the 9th European Space Agency's Earth Explorer Mission, to be launched in 2027. The FORUM core instrument
will be a Fourier Transform Spectrometer (FTS) and it will measure the Earth's upwelling spectral radiance from 100 to
$1600\ \mathrm{cm}^{-1}$ (100–6.25 $\mu$m). FORUM will allow for the first time to observe globally the Earth's spectrally resolved emission
in the FIR.

During the preparatory phase of FORUM, the Far-Infrared Radiation Mobile Observation System (FIRMOS) was employed
to support the mission by validating measurement methods and instrument design concepts, both in the laboratory and in
field campaigns. Throughout this activity, the data gathered have been critically employed for the validation of geophysical
parameters, retrieval codes, and more generally to expand FIR spectroscopic knowledge.

FIRMOS was built at the Italian National Institute of Optics of the National Research Council (INO-CNR), and it was
designed as a laboratory and field campaign flexible instrument. Subsequently it was deployed in the German Alps at the
summit station of the Zugspitze Observatory (2962 m a.m.s.l.) for a two-month campaign (Palchetti et al., 2021) in the winter
2018–19. Some of the measurements collected during that time are presented here to demonstrate the capabilities of the
platform. During the campaign at Zugspitze, FIRMOS was jointly operated with an assortment of co-located instruments that
characterised the observed atmospheric state. The spectra acquired during the campaign were processed to derive higher level
products, namely temperature and water vapour profiles and cloud properties, if applicable.





In this paper we describe the instrument design and its characterisation and discuss the temperature and water vapour products obtained inverting the atmospheric spectral radiance measured during the campaign in clear sky conditions. The retrieval of optical and microphysical cloud properties is the subject of a separate publication (Di Natale et al., 2021). Section 2 introduces and describes in detail the FIRMOS instrument, its optomechanic design, radiometric calibration, electronics and detection specifics; section 3 presents the Level 1 (L1) and Level 2 (L2) data while in section 4 the results are discussed. Finally, in section 5 the conclusions are drawn.

## 2 Materials and Methods

FIRMOS was designed and built first as a laboratory prototype and was successively adapted to obtain a versatile instrument that could be quickly deployed in ground-based field campaigns (<80 Kg, 1 day readiness), specifically at high altitude sites, and easily adaptable to stratospheric balloon flights.

The instrument was built during the compressed schedule preceding the Earth Explorer 9 mission selection and deployed for its first campaign at the Zugspitze Observatory in the Bavarian Alps (South Germany, 47.421°N, 10.986°E, 2962 m a.m.s.l, Palchetti et al. 2021) between the end of 2018 and the beginning of 2019. FIRMOS mostly acquired Atmospheric DLR spectra, in zenith-viewing configuration; at the end of the campaign some days were allocated to surface-looking measurements of a variety of snow samples.

**Table 1.** Characteristics of the measurements performed at Zugspitze during the FIRMOS campaign (Palchetti et al., 2021)

| Type of measurement | Resolution | Integration | Repetition time | Date | No of spectra measured |
|---|---|---|---|---|---|
| DLR spectrum | 0.4 cm$^{-1}$ | 128 s | 256 s | 29 November – 18 December 2018 | 1197 |
| | 0.3 cm$^{-1}$ | 210 s | 420 s | 21 January – 15 February 2019 | 838 |
| Snow and DLR | 0.3 cm$^{-1}$ | 210 s | 420 s | 16 – 20 February 19 | 152 snow + 283 DLR |

A set of instruments was operated in conjunction with FIRMOS: E-AERI, an IR commercial FTS at the Zugspitze summit; a lidar instrument at the Schneefernerhaus station (UFS) at 2675 m a.m.s.l., 700 m to the south-west of the summit station; five dedicated radiosonde launches were carried out from Garmisch-Partenkirchen, 8.6 km to the north-east of the summit. More details are given within the sections below.

### 2.1 The FIRMOS instrument

FIRMOS is a ground-based FTS operating in the far- and mid-infrared range. Its design stems from its predecessor, the Radiation Explorer in the Far InfraRed – Prototype for Applications and Development (Bianchini et al., 2019, REFIR-PAD). The new design, as described in the following sections, is the result of a rationalisation aimed at a leaner instrumental setup and at reducing deployment times by employing commercial parts for motion control and reflective optics.





### 2.1.1 Optomechanics

The optical layout of the FIRMOS interferometer is composed by a double-input and double-output Mach-Zehnder configuration. The setup allows full tilt compensation by employing a movable unit with roof-top mirrors (RTMU). Additional flat

mirrors are used on the right arm of the interferometer to compensate for slit yaw. Parabolic mirrors (45° off-axis) enable light focusing on the detectors, encapsulated with CsI windows. The metrologic source is a 785.9 nm single-mode thermally stabilised laser (Thorlabs). The latter is driven with a constant current from a controller developed in-house, and already employed within the previous REFIR-PAD instrument. The reference laser follows the same optical path as the infrared beam, with dedicated optics joined to the same mountings as for the main measurement.

Radiometric accuracy is achieved by employing three blackbody source, the hot (HBB) and the cold (CBB) calibration blackbodies and the reference blackbody (RBB). A rotating mirror (PM0) located at the first interferometer input can select either the HBB, the CBB or the sample scene (Figure 1), the contribution of this mirror to the instrument response is therefore accounted for in the calibration procedure (see Section 2.1.3). The RBB, located at the second input, is in thermal equilibrium with the other optical components. At every measurement cycle, the calibration procedure is performed before and after the

sample scene.

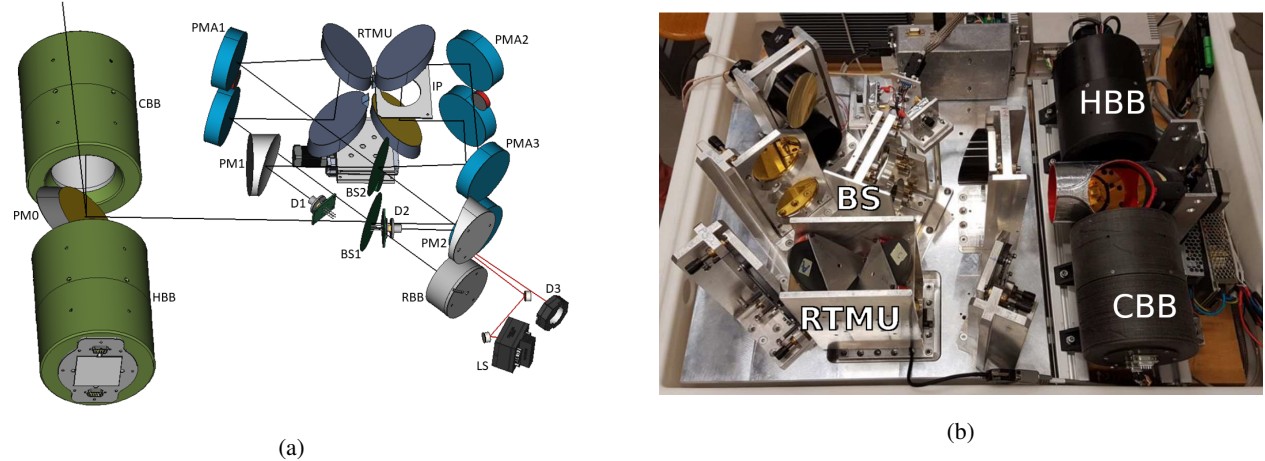

(a)           (b)

**Figure 1.** FIRMOS: (a) optical layout diagram, the blackbodies (HBB and CBB) are depicted in green, PM0 indicates the scene selection mirror. The roof-top mirrors unit (RTMU) is on the top right, IP is the internal pupil, in the centre BS1 and BS2 indicate the beam splitters. The whole optical path is folded on two levels using mirrors (PMA1, PMA2, PMA3, PM1, PM2). Also shown are the pyroectric detectors (D1 and D2) the metrology laser (LS) and its detector (D3), the reference (RBB) (b) picture of the inner structure of the instrument

The FIRMOS setup was designed to maximise the optical throughput by employing 76.2 mm diameter optics while maintaining a field of view of 22 mrad. In addition, the optical system is image-forming at the detector, although the latter is a single pixel (diameter 2 mm). The above features are meant to enhance the observed scene selectivity while maintaining good signal-to-noise ratio, and therefore to facilitate the development of software tools for geophysical parameters retrieval.





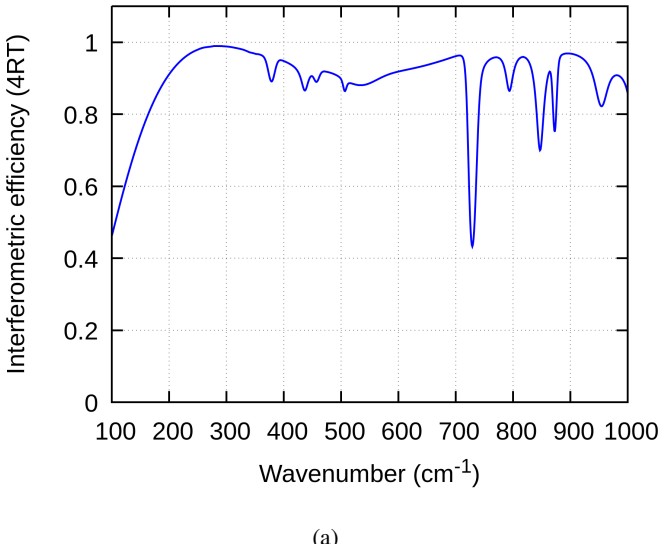

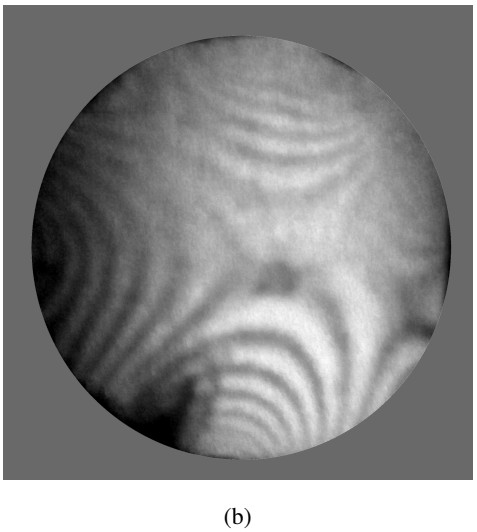

(a)

(b)

**Figure 2.** (a) Interferometric efficiency of the beam splitter. (b) Interferometric image of a beam splitter sample.

The field of view in FIRMOS is defined by the optical path length of the instrument, the internal pupil radius, and the detector area, the latter being the main limiting factor in the current design. The optical specifications of the instrument are listed in Table 2.

**Table 2.** Optical collection specifications.

| Spectral Coverage | 100–1000 cm$^{-1}$ |
|---|---|
| Maximum Spectral Resolution | 0.25 cm$^{-1}$ |
| Optical throughput | 0.0063 cm$^2$ sr |
| Beam Aperture (Field of view) | 22.4 mrad |
| Internal pupil at RTMU | 45 mm diameter |
| Internal optical path length | 1425 mm |

To cover the IR spectral range from 100 cm$^{-1}$ to 1000 cm$^{-1}$, the instrument adopts wideband Germanium-coated Biaxially-oriented PolyEthylene Terephthalate (BoPET) beam splitters (BS) and room temperature deuterated L-alanine doped Triglycene

Sulphate (DLATGS) pyroelectric detectors. The absorption of the BoPET BS substrate causes some degradation in efficiency in some narrow bands around 730, 850, 873 and 973 cm$^{-1}$, as it can be seen in Figure 2 which shows the typical 4RT efficiency. The instrument spectral range is limited at low wavenumbers by the absorbance of the detector CsI windows and, at high wavenumbers by degradation of the optical performance due to BSs flatness errors (see Fig. 2(b) and Fig. 6).

The BS samples were manufactured at INO-CNR and tested with a Newton interferometer, to select those with maximum

flatness. The interferometer is capable of detecting flatness anomalies with 0.1 $\mu$m precision by employing a reference surface





with a flatness of $\lambda/20$, a monochromatic source and a digital camera. The pattern observed in the case of membranes with a divergence from flatness of a few micrometres is of the saddle or multi-saddle type, especially close to the edge. The saddle peak-valley distance is evaluated through the measurements of the number of fringes on the main saddle along a track (Fig.2(b)). The best two BS samples, with flatness error of less than 2.5 $\mu$m peak-valley, were integrated on FIRMOS to guarantee good performance over the 100–1000 cm$^{-1}$range.

A lightweight and compact linear stage model (Zaber model X-LSM025A, mass <0.5 Kg, height 20 mm, centred load capacity 100N) was used, installed within a notch of the breadboard below the RTMU to perform the interferometric scan. A scanning speed of 0.25 mm/s in a 30–60 s acquisition time for a single scan is used. The typical standard deviation of speed over a scan was obtained experimentally as 0.043 mm/s at a 0.25 mm/s scan speed, sufficiently stable to be accounted for during the signal analysis.

The RTMU was manufactured from a monolithic aluminium piece (see Figure 1(b)). The mirrors are placed in a roof-top configuration and fixed by a system of springs and screws.

The instrument was designed for easy transportation and deployment. Its size is 85x95x50 cm, it weighs 80 kg, and the power consumption is 60 W. A plastic enclosure was used to protect against environmental conditions, an 8 cm diameter aperture with a motorised shutter was used for observation.

The instrument breadboard was realised as a monolithic aluminium slab with a mass of 17.5 Kg, and dimension of 520 x 540 x 45 mm (L x W x H). Rods spacing and tightening points were initially designed balancing dimensions, mass and stiffness of the framework. The final layout was identified through an iterative design process carried out with CAD software that evaluated static loads.

### 2.1.2 Radiometric Calibration Unit

In order to perform a calibrated radiometric measurement, at least two known radiation sources are required. In FIRMOS the HBB and a CBB are located at the instrument entrance. The scene mirror allows the acquisition of the external view of the instrument (Zenith or Nadir) or one of the two BBs. The axial rotation is obtained through a stepper motor (NEMA 17 stepper), that also supports the mirror, surrounded by a plastic guard in order to prevent stray light from other instrument components. The support was assembled out of 3D-printed high strength co-polyester plastic parts.

Montecarlo numerical calculations were performed to optimise the cavity geometry of BBs, in order to maximise normal emissivity, a 34 ° angle was chosen for both the HBB and CBB inner cones Palchetti et al. (2008).

The CBB was assembled in a 3D-printed co-polyester plastic shell and the HBB was assembled in a 3D-printed heat resistant carbon fiber reinforced Nylon plastic shell. They were both designed to minimise thermal dispersion and were coated using NEXTEL-Velvet-Coating 811-21. Some layers of thermal superinsulation foils were placed inside the plastic shells, in order to minimise the thermal exchange between the BBs and its supports.

The BBs controllers are two modular drivers for temperature reading and stabilisation developed in-house. The temperature of the RBB is monitored by a supplementary module of the HBB driver. Each BB controller simultaneously records the temperature of four sensors: one high-accuracy (30 mK) PT100 sensor, used for temperature reading; one high-resolution (500





μK) NTC sensor, for active temperature stabilisation; and two one-wire digital thermometers (Dallas DS18B20), placed at the opposite extremities of the BB, in order to check the BB temperature homogeneity. A comparison between the PT100 temperature reading by the FIRMOS controller and by a commercial Temperature Monitor (Lakeshore, Model 218) with accuracy of 0.6%, showed a positive offset of 200 mK between the controller and the Lakeshore sensor, which was subtracted during the signal analysis.

For the field campaign, the CBB and the HBB were typically stabilised at a temperature of 15°C and 60°C, respectively. Temperature control performance results are shown in Table 3. In order to test the BB thermal stability, the temperature of the PT100 sensor was recorded after the switching of the stabilisation. The difference between the PT100 reading and the stabilisation temperature of each BB is reported in Figure 3 (a)-(d).

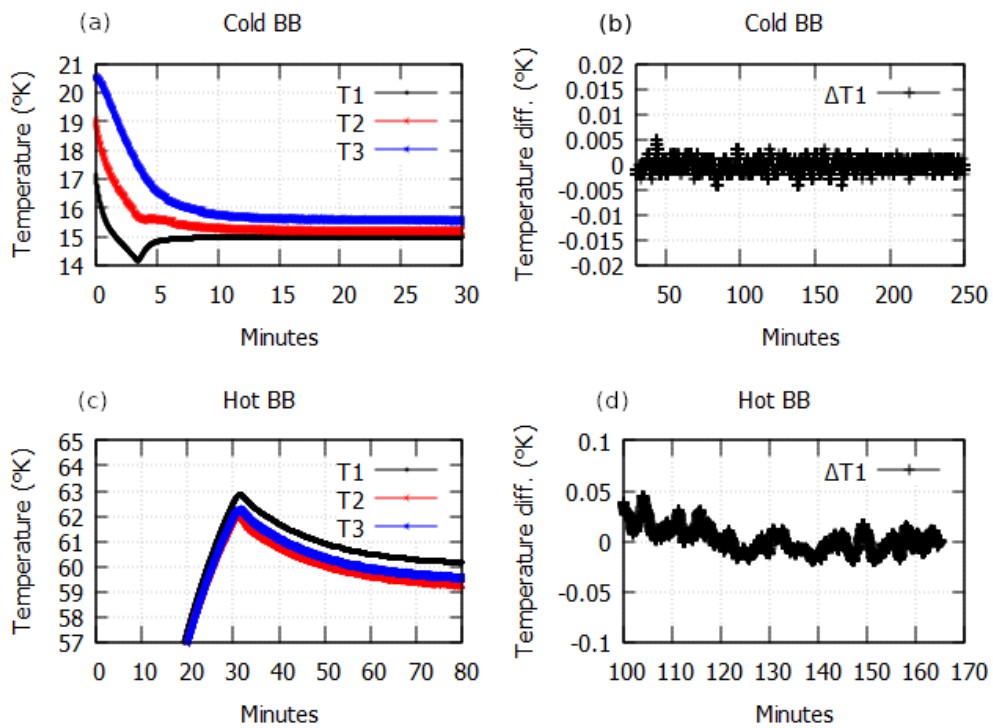

**Figure 3.** Thermal stabilisation time of (a) Cold BB and (c) Hot BB. Temperature differences from average value at longer times are also reported for (b) CBB and (d) HBB, for each sensor (T1: PT100, T2 and T3: DS60B18).

The CBB reaches the stabilisation temperature in less than 1 hour and the HBB in about 2 hours. For the CBB the standard
deviation was calculated for 3 hours and 45 minutes, starting 30 minutes after the switching of the stabilisation (Figure 3(b)). For HBB, the standard deviation was calculated for approximately 45 minutes, beginning 2 hours after the switching of the stabilisation (Figure 3(d)). The BB temperature homogeneity is inferred by registering the temporal evolution of the difference between the readings of the two Dallas sensors. After one hour, the thermal gradient of both BBs is approximately 0.3 K.





**Table 3.** BB temperature control results

|                         | HBB      | CBB      |
|-------------------------|----------|----------|
| Working Temperature     | 60 °C    | 15 °C    |
| Stabilisation Precision | 8.3 mK   | 1.1 mK   |
| Stabilization Accuracy  | 30 mK    | 30 mK    |
| Thermal gradient        | 0.3 K    | 0.3 K    |

### 2.1.3 Detectors and Electronics

One of FIRMOS enabling technologies is the adoption of two room-temperature pyroelectric DLATGS detectors covering the mid-infrared as well as the far-infrared region. The detectors are uncooled (model: Selex P5180) and have a noise equivalent power $NEP \equiv \sqrt{A}/D^*$ of 1.4 and 1.6 $10^{-10}W/\sqrt{Hz}$, where $A$ and $D^*$ are, respectively, the detector area (3.14 mm$^2$) and the detectivity. The pyroelectric preamplifiers were prepared at INO-CNR and the electric scheme follows a classic design, previously tested for the REFIR-PAD instrument (Bianchini et al., 2019). The original scheme was optimised miniaturising as 160 much as possible the amplifier to reduce the wiring length, in order to increase immunity to electromagnetic interference noise.

The slow response of pyroelectric detectors requires to compensate for the acquired signals with digital processing in order to remove amplitude and phase distortions. For this purpose, the frequency response of the detector and of the pre-amplifier subsystem were characterised, measuring their frequency response to a laser-beam step excitation for both output channels (Fig. 4). An empirical model was successively derived from the measurements with a fitting procedure and then used to digitally com-165 pensate for the detector response during the L1a analysis (described in Section 3.1).

The FIRMOS detectors observe signal variations in the range of 5–100 Hz depending on the scanning conditions.

## 3 Data analysis

### 3.1 Level 1 data analysis (spectral calibration)

The L1 data analysis processes the interferograms acquired by the instrument to obtain calibrated spectra. The procedure 170 follows the one described in more detail in Bianchini et al. (2008) for a double-input/double-output ports interferometer and is divided into 3 steps:

- L1a performs the signal conditioning (filtering, detector response compensation, path-difference resampling, phase correction, etc.) and the Fourier transform;

- L1b carries out the radiometric calibration providing the calibration functions and the calibrated spectra for each output 175 channel;

- L1c calculates the average spectrum for every measurement cycle, composed of sky observations and calibration measurements. L1c provides one average spectrum for each of the two output channels, as well as the average of the two





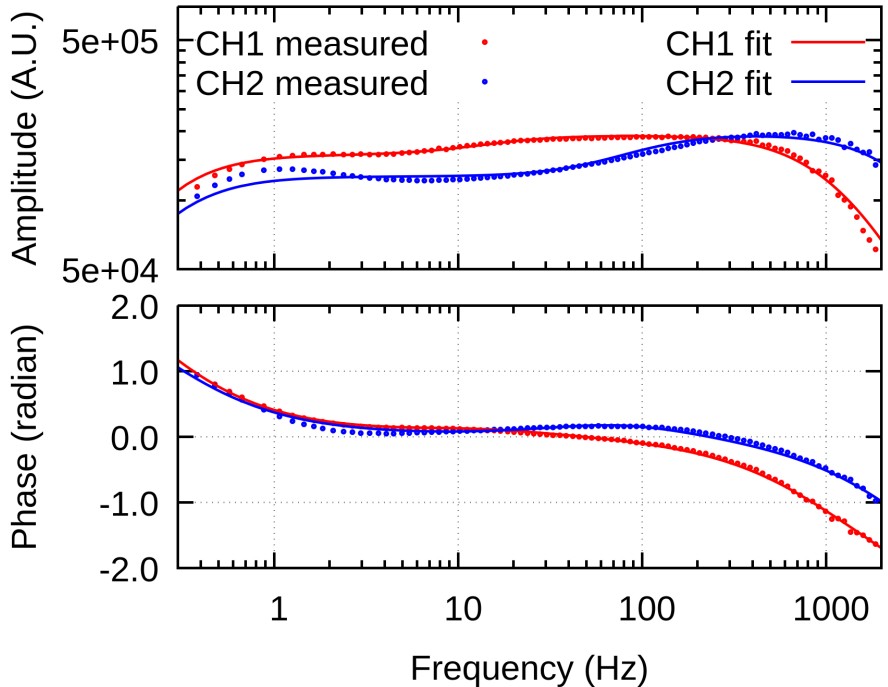

**Figure 4.** Frequency response of the detection system to a laser step excitation.

channels together with an estimate of the noise and the calibration error. All the averages are weighted by the respective noise estimate.

180     Each of the interferometer output signal is proportional to the difference of the two input signals with a wavenumber-dependent complex response function F, that is, in general, different for the two inputs, as well as for the two output channels. As described above, the first input is used to measure the scene, whereas the second input, which corresponds to the instrument self-emission, looks continuously to the RBB source. Under these conditions, the relationship between the uncalibrated complex spectrum $S(\sigma)$, and the calibrated spectrum of the observed scene $L(\sigma)$, for each output channel, is given by the following.

185     $$S(\sigma) = F_1(\sigma)L(\sigma) - F_2(\sigma)B_r(\sigma) \tag{1}$$

where $F_1$ and $F_2$ are the calibration functions and $B_r(\sigma)$ is the radiance from RBB, calculated from its measured temperature using the Planck law.

Calibration is carried out by changing the observed scene with the rotating mirror at the first input. The calibration functions $F_1$ and $F_2$ are obtained from a two-point radiometric calibration procedure, measuring sequentially the radiance of the HBB



and CBB during each measurement cycle. The calibrated radiance spectrum $L(\sigma)$ is then calculated from the uncalibrated spectrum $S(\sigma)$ and the theoretical expression of $B_r(\sigma)$:

$$L(\sigma) = \Re \left\{ \frac{S(\sigma)}{F_1(\sigma)} + \frac{F_2(\sigma)}{F_1(\sigma)} B_r(\sigma) \right\} \tag{2}$$

As noted in Bianchini et al. (2008) all the quantities used in the calibration procedure, are complex, only in the last expression, Eq. 2, the real part of the result is taken, obtaining the measured spectrum as a real quantity. Furthermore, since the optical layout of the interferometer is equivalent with respect to the two inputs, $F_1$ and $F_2$ have almost the same values. Forward and reverse sweeps of the interferometer (optical path difference, $OPD : -OPD_{max} \rightarrow +OPD_{max}$ and $OPD : +OPD_{max} \rightarrow -OPD_{max}$) are treated separately during the calibration, since in general they will have different phase errors, nonetheless, the final spectral radiances can be averaged.

The precision of each measurement is calculated in terms of the noise equivalent spectral radiance (NESR) that has to be associated with the specific observation. This quantity depends on the number of acquisitions of the observed scene and the number of HBB/CBB calibration measurements during each measurement cycle, and it is dominated by the detector noise (random error component) $\Delta S$, which is independent of the observed scene. The NESR, is then obtained through error propagation of $\Delta S$ on the calibrated spectrum obtaining:

$$NESR = \frac{\Delta S}{F} \sqrt{\frac{1}{N} + \frac{2}{n} \left( \frac{\bar{S}}{\bar{S}_H - \bar{S}_C} \right)^2} \tag{3}$$

where $\bar{S}$ is the average of $N$ scene acquisitions (four in FIRMOS standard acquisition configuration), $\bar{S}_h$, $\bar{S}_c$ are the averages of $n$ HBB and CBB acquisitions (2 in standard configuration), respectively. $F_1$ and $F_2$ are considered equal to F for the noise calculation. $\Delta S$ is obtained from the standard deviation of a series of uncalibrated measurements of a constant source, such as the CBB. The spectral dependence of all the variables in Eq. 3 is omitted for the sake of brevity.

Figure 5 reports the results for a typical observation of the atmosphere (NESR_atm) and of a reference blackbody source, measured inside the laboratory (NESR_bb). The NESR has sharp spectral features, where the noise increases, due to the absorption of the gases inside the interferometric path, mainly water vapour below $400\ \mathrm{cm}^{-1}$ and carbon dioxide at $667\ \mathrm{cm}^{-1}$, and the absorption bands of the BoPET BS, around 730, 850, 873 and $973\ \mathrm{cm}^{-1}$. Furthermore, the NESR estimate depends on the observed scene because of the error on the calibration source measurements that propagates on the NESR estimate through the calibration functions. If numerous calibration measurements are performed so that n is large enough to neglect the second term in Eq. 3 compared to 1/N, then the NESR estimate does not depend anymore on the observed scene and becomes an instrument specification, see NESR_instr curve in the bottom panel of Fig. 5. The latter approach is typically applied to specify the instrument performance in terms of the NESR, whereas the second term of Eq. 3 is accounted for in the calibration precision. However, in our case, we perform a calibration for each measurement cycle so that n is comparable with N, therefore, the total NESR estimate of Eq. 3 is a better estimate of the total random error of our single measurement.





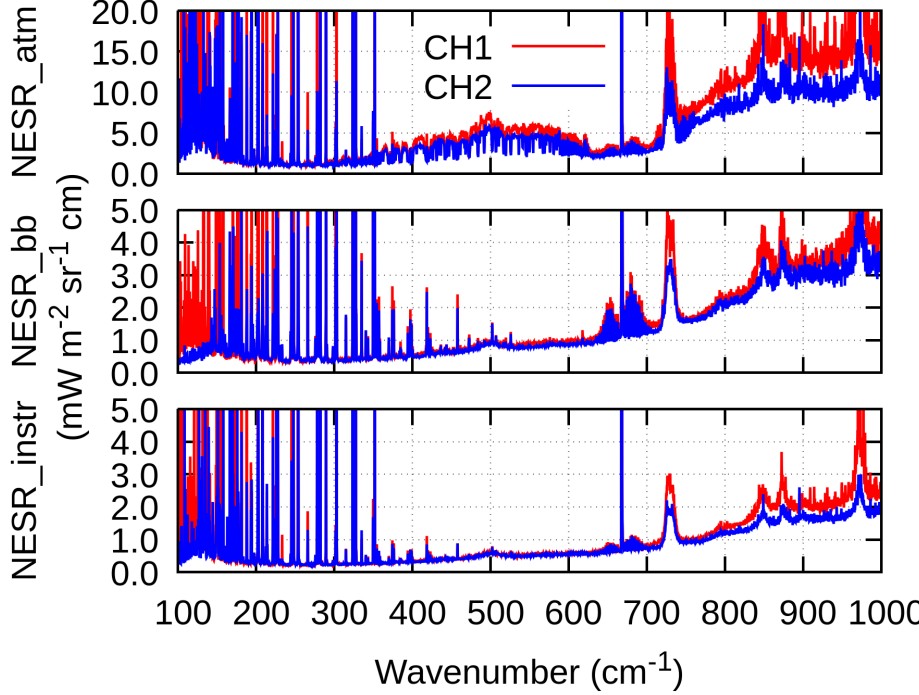

**Figure 5.** NESR on the calibrated spectra calculated from error propagation of Eq.3, in the case of a measurement cycle of $N = 4$ sky measurements and $n = 2$ for each calibration sources. NESR_atm (upper panel) is for the observation of the atmosphere in clear sky condition, NESR_bb (middle panel) is for the observation of a blackbody souces in laboratory, and NESR_instr (lower panel) is the instrumental component equal to $\Delta S / F * \sqrt{(1/N)}$

.

The increase of noise over $600\,\mathrm{cm}^{-1}$ in Figure 5 indicates a performance degradation. Such degradation is mainly caused by the BS flatness error (see Fig. 2), as it can be inferred by Figure 6 that shows the comparison of the measured NESR_instr (the same curves of the bottom panel of Fig. 5) with simulations carried out assuming a simple numerical model of the instrument NESR. The model includes the detector specifications and the optical efficiency of the interferometer, in the simulations the interfering wave fronts were distorted with a spherical shape to approximate the BS flatness error. The results of Fig. 6 demonstrate that measurements are consistent with a BS flatness error of about 2.2 μm in accordance with the results shown Fig. 2.

The calibration accuracy is dominated by the accuracy of the blackbodies' temperature measurement. The corresponding calibration error CalErr is spectrally correlated but independent from one measurement to another and is obtained through the error propagation of the temperature accuracy measured on each reference blackbody, which is conservatively assumed to be equal to 0.3 K. Taking into account the measurement error on the temperature of each blackbody as independent, we can calculate the corresponding uncertainty on the theoretical Planck emission as $\Delta B_H$, $\Delta B_C$, and $\Delta B_R$ and the resulting



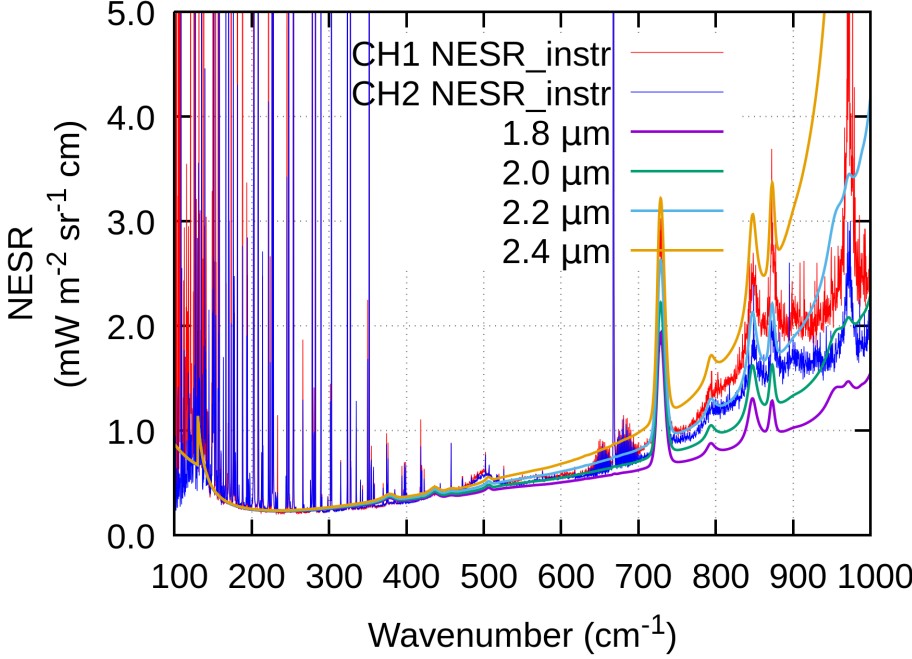

**Figure 6.** Comparison of the measured NESR_instr (red and blue curves) with simulated NESR obtained with different values peak-valley of BS flatness errors shown in the legend.

calibration error through the error propagation in Eq. 2 obtaining the following:

$$CalErr = \sqrt{\Delta B_R^2 + \left(\frac{\bar{S}}{\bar{S}_H - \bar{S}_C}\right)^2 (\Delta B_H^2 + \Delta B_C^2)} \tag{4}$$

As shown in Fig. 7, the calibration error estimate also depends on the observed scenes and is larger for colder scenes when the sky is observed, since the uncalibrated signal S, which depends on the temperature difference between the observed scene and RBB (see Eq. 1) is larger, in this case.

Finally, in Fig. 8 is shown an example of the spectrum and error estimates obtained as a weighed mean of the two channels after the L1c analysis. The measurement was acquired in clear sky conditions during the campaign at Mount Zugspitze in measurement cycles, each comprised of four sky and four calibration observations, as described above. The total sky observation has a duration of 215 s and the total measurement cycle time is eight minutes. The standard deviation (STD in the figure) of the measurement, which is in good agreement with the NESR estimate used in the mean, is also shown in the figure.





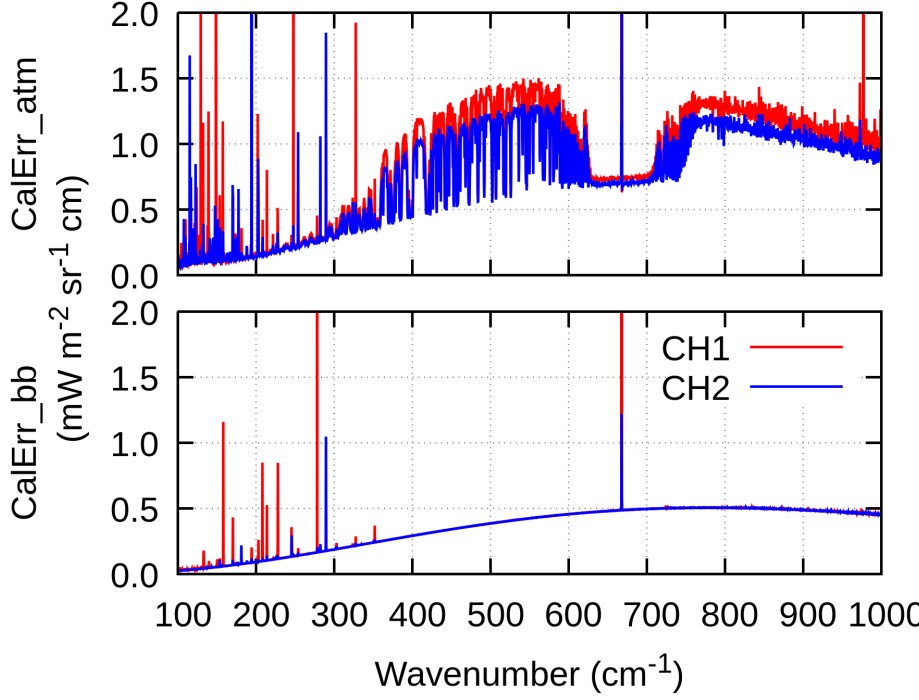

**Figure 7.** Calibration error calculated from a conservative estimation of 0.3 K uncertainty on the temperature measurement of each reference blackbody. CalErr_atm (upper panel) is for the observation of the atmosphere in clear sky condition, CalErr_bb (lower panel) is for the observation of a blackbody souce in laboratory.

### 3.2 Level 2 data analysis (retrieval)

FIRMOS L1 measurements were processed using the Kyoto protocol and Informed Management of the Adaptation (KLIMA) forward and retrieval models (Sgheri et al., 2021; Ridolfi et al., 2020; Del Bianco et al., 2013; Bianchini et al., 2008; Carli et al.,
2007) to derive geophysical products (L2). Only spectra acquired from the instrument channel one were used since the second channel occasionally showed a degradation that could have a negative impact on the results. The retrieval of water vapour and temperature profiles was carried out on the entire clear sky L1 dataset (as defined in Section 3.3), in the range of 200 cm$^{-1}$ to 1000 cm$^{-1}$. The targets were retrieved from the surface up to 7 km on seven atmospheric layers for temperature and 6 for water vapour.

The algorithm uses an optimal estimation approach (Rodgers, 2004) and a multi-target retrieval strategy (Carlotti et al., 2006). Profiles from the National Centers for Environmental Prediction (NCEP) reanalysis (Kanamitsu et al., 2002) were used as initial guess and a-priori. The a-priori errors on temperature and water vapour were set to respectively 0.3% and 50% of the averaged a-priori values.





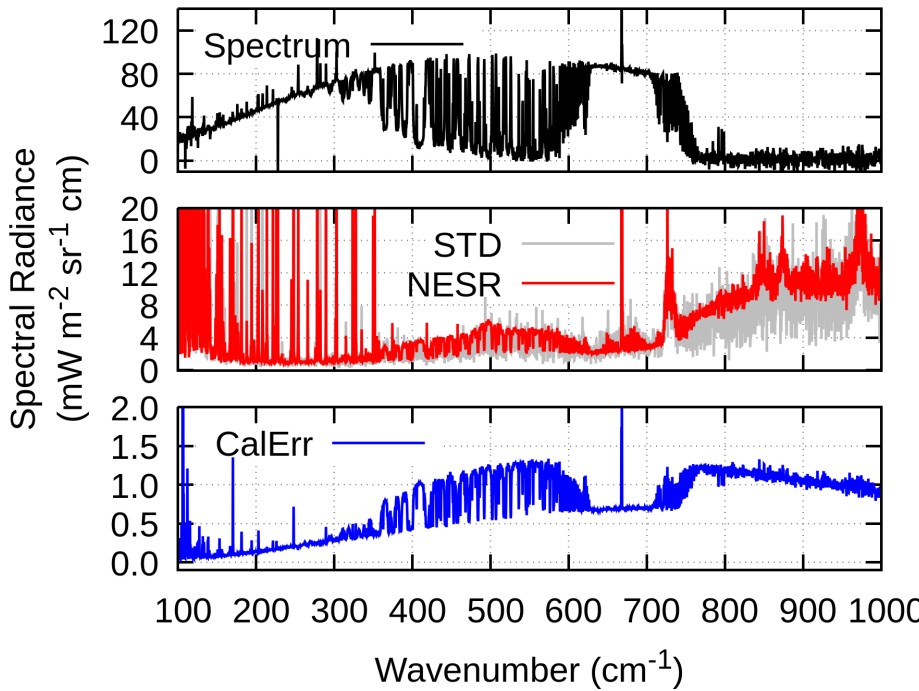

**Figure 8.** Calibrated spectrum (upper panel) and error estimates (NESR and STD middle, Calibration error lower panel) obtained from the weighted mean of the two output channels in a measurement cycle of four zenith observations in clear sky conditions on 25 January 2019 at 15:55 UTC.

### 3.3 Clear-sky selection criteria

The KLIMA model can analyse pure clear sky scenes as well as scenes with very optically thin clouds, for this reason a subset of measurements not significantly perturbed by clouds in the FIRMOS band was first selected. The subset is referred to as the *clear-sky cases* subset.

Clear-sky cases were selected by evaluating the transparency and slope (gradient) of the FIRMOS spectra in the Atmospheric Window (AW, 820–980 $cm^{-1}$). In absence of clouds, the spectrum in the AW is well known and equal to the contribution of the 260 water vapour continuum, which is small in comparison to the measurement noise. Likewise, an ideal noise-free and cloud-free measurement would have a gradient of 0, whereas negative values would correspond to a noise-free cloudy observation with a magnitude depending on the specific cloud.

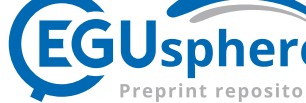



The transparency of the AW was assessed in the narrow spectral window between 829-839 $\text{cm}^{-1}$ by calculating the spectral average of the ratio of the signal with respect to the total noise as follows:

$$\Delta = \frac{1}{\nu_2 - \nu_1} \int_{\nu_1}^{\nu_2} \frac{S(\nu)}{\sqrt{\text{NESR}^2(\nu) + \text{CalErr}^2(\nu)}} d\nu \qquad (5)$$

where $\nu_1$ and $\nu_2$ are the extremes of the AW spectral range, S the measured spectral radiance and the quadratic sum of NESR and calibration error constitutes the total noise. The absolute value of $\Delta$ for a clear sky observation is expected to be less than one, indicating that the measured signal is only due to noise fluctuations.

The slope was calculated between 786 and 961 $\text{cm}^{-1}$ in 6 specific microwindows where gas absorption lines are absent:
(786–790, 830–835, 856–863, 893–905, 912–918, 960–961 $\text{cm}^{-1}$). The microwindows were selected from a spectrum simulated by the KLIMA forward model; the gradient was obtained from a linear fit of radiance on the microwindows. The fitted slope showed maximum positive values up to $5 \cdot 10^{-5}$. The latter are not physically consistent, and they can be related to noise fluctuations around zero.

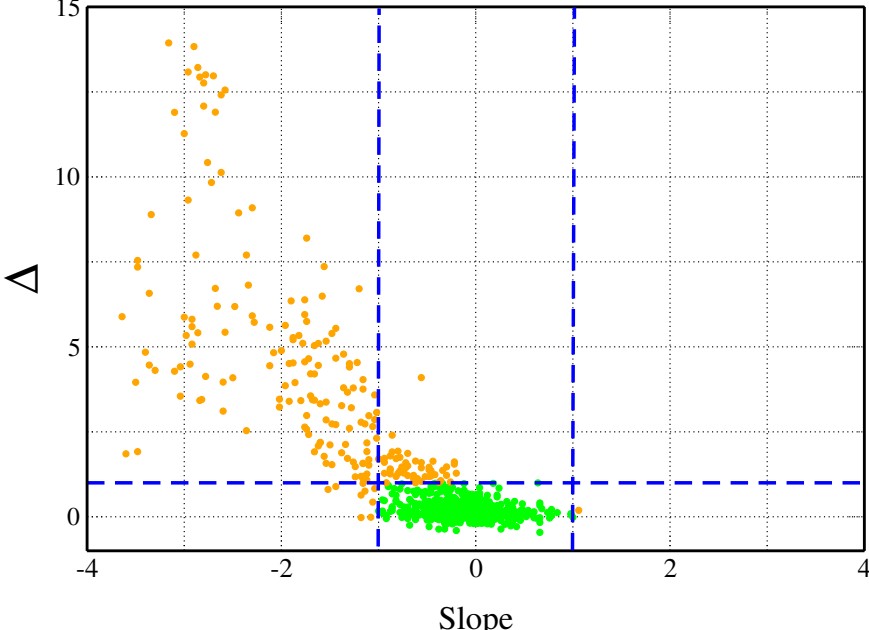

**Figure 9.** Plot of the $\Delta$ ratio, as defined in Eq. 5, versus the normalised slope calculated using the 6 microwindows in the spectral range 786-961 $\text{cm}^{-1}$ defined in the text. Blue dashed lines indicate the acceptance values and the green and orange dots denote the spectra analysable and not analysable with KLIMA, respectively.

In Figure 9, slope values, normalised with respect to their maximum, are plotted (abscissa) against $\Delta$. The condition $\Delta < 1$
corresponds to slope values within the range (-1, 1), except for a few negative cases. We assumed that spectra laying between



the thresholds, defined by the dashed blue lines, represent the set of measurements which can be analysed by the KLIMA code. This is a reasonable choice since, as long as the signal is lower than the instrumental noise, the slope in the atmospheric window lies within the range $[-5 \cdot 10^{-5}, 5 \cdot 10^{-5}]$ with a symmetric distribution around 0. Of a total of 838 spectra, 625 (green dots in Figure 9) fell within the acceptance region (blue dashed lines) and were therefore analysed with KLIMA, 213 spectra
(orange dots) were discarded.

## 4 Results

### 4.1 Retrieval of geophysical paramenters

The high number of measured spectra (625, clear sky) allowed a statistical analysis of the retrieval results. In particular, it is important to assess the quality of the retrievals by analysing the $\chi^2$ distribution. Figure 10 shows the reduced $\chi^2$ distribution,
and a clear minimum is found for the value of $\chi^2 = 1.2$ (red line). With this criterion, 60 out of 625 measurements of the clear sky selection were excluded. This threshold was verified being a conservative choice, as it guarantees the exclusion of all problematic L1 FIRMOS measurements.

The maximum number of occurrences of the distribution lies between 0.6 and 0.7, indicating a probable overestimation of the NESR of the FIRMOS instrument of about 25% on average. The time series of the final reduced $\chi^2$ obtained from the
fitting procedure is shown in Figure 11, where the red line indicates the threshold value.

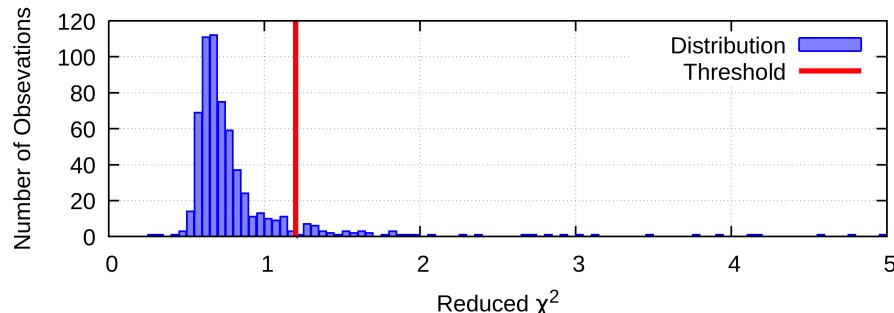

**Figure 10.** Distribution of the reduced $\chi^2$ using a bin width of 0.05. The red vertical line at 1.2 indicates the threshold corresponding to an evident minimum (close to zero cases) of the distribution.

Measurements that satisfy the acceptance criterion $\chi^2$ were used for a statistical analysis of the residuals. The latter are calculated as the difference between the simulated spectrum at the last iteration of the retrieval and the FIRMOS observation. The mean and standard deviation of residuals provide an a-posteriori estimation of the measurements' calibration error and NESR, respectively.
Figure 12 compares the standard deviation of the residuals (blue line) to the average NESR (red line). The residuals' standard deviation curve correctly reproduces the shape of the average NESR curve of the FIRMOS measurements. However, as



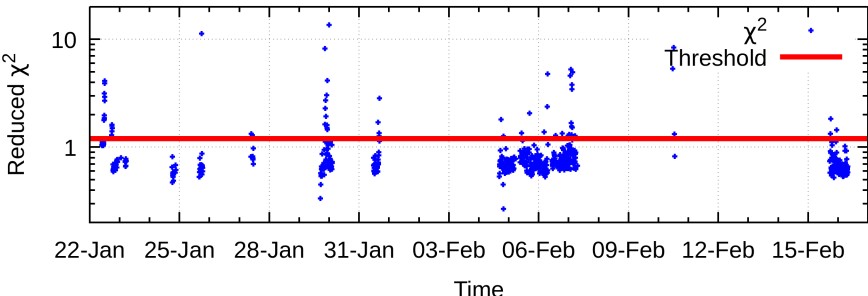

**Figure 11.** Time series of the reduced $\chi^2$ obtained from the fitting procedure. Red line indicates the threshold at $\chi^2 = 1.2$ defined in Figure 10.

observed for the reduced $\chi^2$ distribution, the values of the curves indicate a probable overestimation, on average by 25%, of the NESR of the FIRMOS measurements. The same NESR reduced by 25% is also shown in green.

Figure 13 shows the comparison between the average of residuals (blue line) and the averaged calibration error (red line). The grey shading is the average NESR divided by the square root of the number of observations (the standard error of the mean). In this case, both the calibration error and the residual NESR are quantitatively consistent with the average of the residuals.

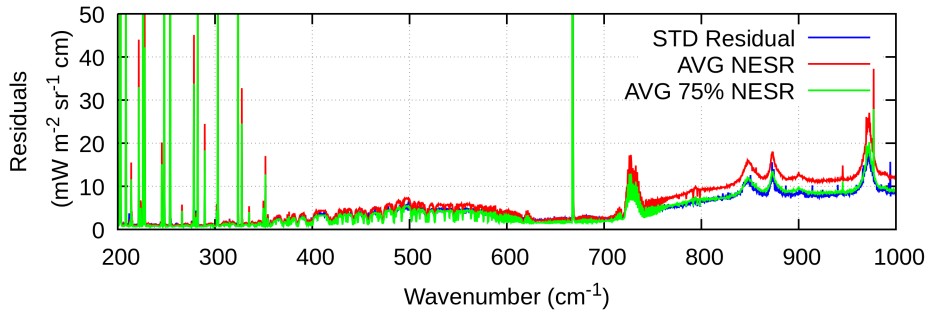

**Figure 12.** Comparison between the standard deviation of the residuals (blue line), the averaged NESR (red line), and the averaged NESR reduced by 25% (green line).

The vertical distributions of water vapour and temperature were retrieved from FIRMOS observations for 6 and 7 atmospheric levels, respectively (Figures 16 and 17), from the surface up to 7 km. The time series of the Degrees of freedom (DOFs) (Rodgers, 2004) for water vapour (green points) and temperature (red points) profiles are shown in Figure 14. Within the FIRMOS measurements, we observe strong variability of the information content. In particular, water vapour shows variations from 2 to 4.5 DOFs. The latter is associated to larger water vapour content near the surface and therefore to the instrument. As an example, we consider two results with high and low DOFs, respectively. In Figure 15 water vapour (left) and temperature (right) retrieved profiles are shown respectively for the two cases under consideration. A larger water vapour content is shown



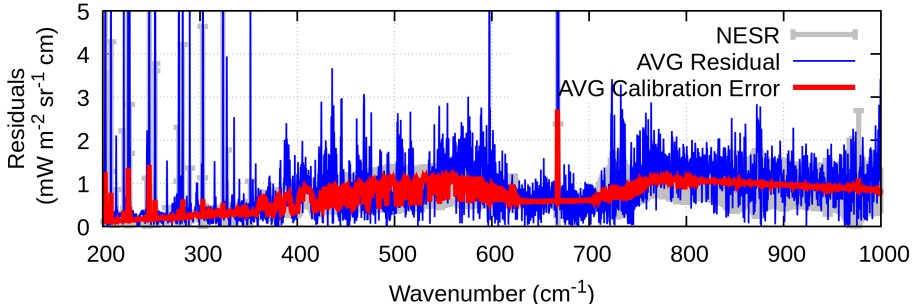

**Figure 13.** Comparison between the average of the residuals (blue line) and the averaged calibration error (red line). The grey shading is the residual NESR after the average.

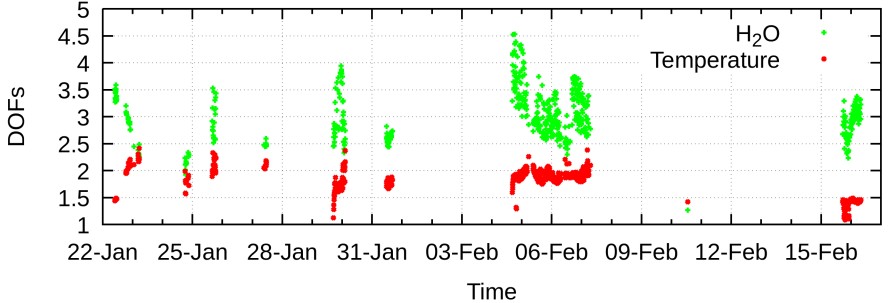

**Figure 14.** Time series of the DOFs of water vapour (green) and temperature (red) profiles obtained from the FIRMOS observations.

near the surface for the low DOFs case (red curve). The obtained retrieval errors are also larger when the surface water vapour content is higher. Instead, temperature profiles do not show relevant variations.

Figures 16 and 17 show the Averaging Kernel profiles (Rodgers, 2004) for water vapour and temperature, respectively. Retrieved profiles were obtained from two FIRMOS measurements with low (on the left) and high (on the right) water vapour content. The vertical resolution profile is also shown (red dashed line). The names of the retrieved species, the total DOFs

of the target species, and the number of fitted points are shown in the inset of the figure. High water vapour content in the atmosphere reduces the retrieval DOFs, deteriorating the vertical resolution. Instead, when the effect of water vapour content on temperature retrieval is less significant, both the Averaging Kernel profiles and the vertical resolution show little variation.

## 4.2 Comparisons

The water vapour and temperature profiles retrieved from FIRMOS spectra were compared with those provided by the ra-

diosoundings, those retrieved from the Raman lidar (only water vapour) and with the ERA5 reanalysis products.



**Figure 15.** (a) Retrieved profiles of the water vapour mixing ratio and (b) temperature. Error bars correspond to retrieval errors. The profiles were obtained from two FIRMOS measurements with high (blue curves) and low (red curves) information content. For water vapour, the DOFs are 4.18 and 2.69 respectively, for temperature 2 and 1.78 as also shown in Figures 16 and 17.

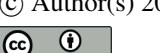



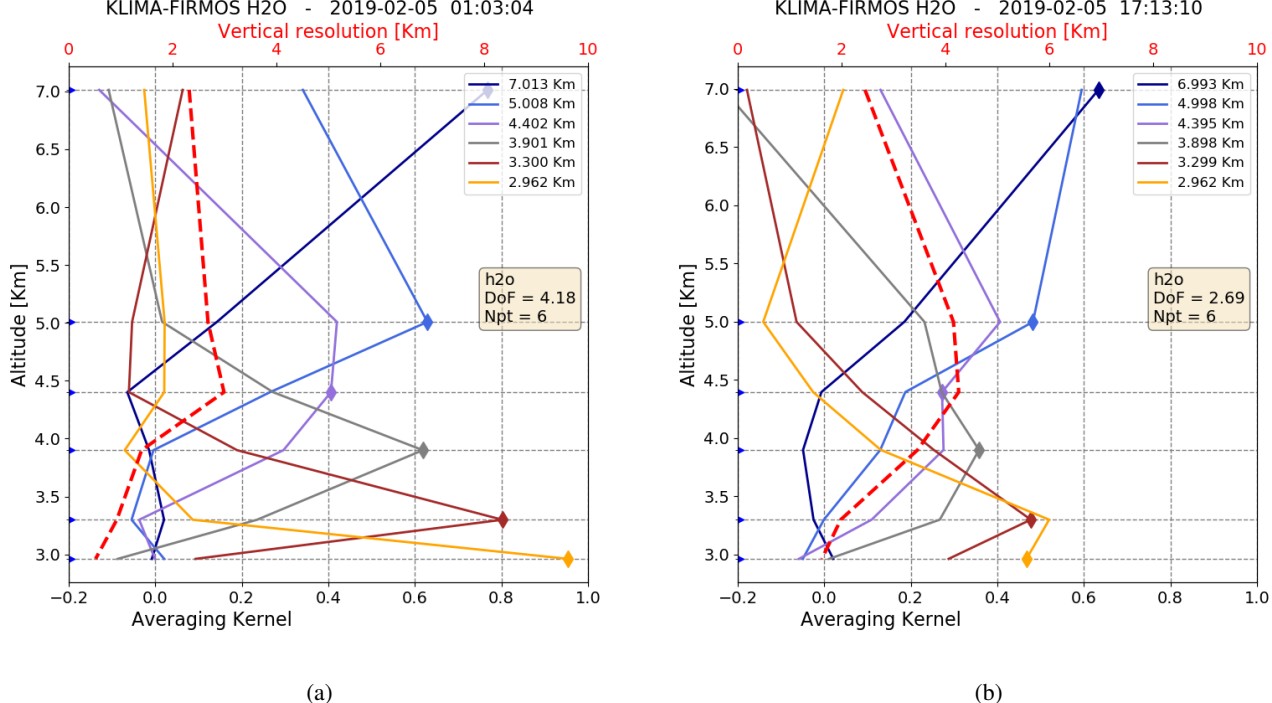

(a)                                                                                         (b)

**Figure 16.** Averaging Kernel profiles (continuous curves) related to retrieved water vapour profiles as obtained from two FIRMOS measurements with high (a) and low (b) information content. The vertical resolution profile is also reported (red dashed line). (inset) Total DOFs and number of fitted points.

### 4.2.1 Comparison with radiosonde measurements

Five dedicated balloon launches were carried out by a team from the Forschungszentrum Jülich at the Institut für Meteorologie und Klimaforschung (IMK-IFU, part of the Karlsruher Institut für Technologie, KIT) in Garmisch-Partenkirchen, 8.6 km northeast of the summit. The balloons were launched at 18:03, 19:03, 23:00 CET on 5 February and at 18:33 and 23:33 CET the

following day.

Air temperature and water vapour mixing ratio from standard Vaisala RS41-SGP radiosondes were compared to the three individual FIRMOS L2 data nearest in time, in order to evaluate the retrieval products quality. Table 4 lists the measurement time of the FIRMOS data used in the comparison and the corresponding balloon launch.

The RS41 temperature measurement has accuracy 0.3 K and precision 0.15 K, the humidity sensor accuracy is 10%, precision

2%, the quality of the radiosonde water vapour measurements were checked with an accompanied high accurate frostpoint hygrometer (CFH, for details see Palchetti et al., 2021).

Figure 18 shows the radiosonde flight trajectories while their altitude was between 3 and 10 km. The radiosoundings launched on 6 February were under thin cirrus cloud conditions and much farther from Zugspitze, so they were not included in the




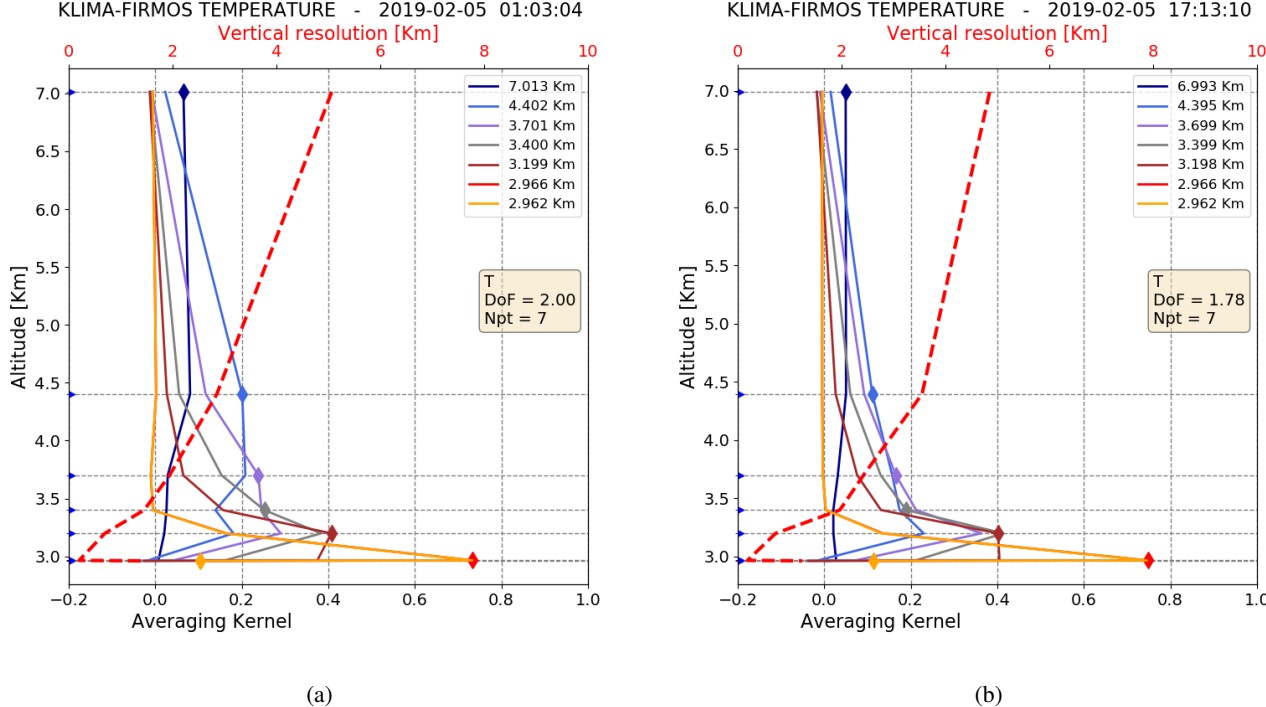

(a)                                                           (b)

**Figure 17.** As in Figure 16 for the temperature profiles.

**Table 4.** Radiosondes launches for 5 February 2019, and corresponding FIRMOS measurements used in the comparison. The central column specifies the time at which the sonde reached the altitude at which FIRMOS was located. All the times are given in CET time.

| Launch time | time @2957 m | FIRMOS measurement time |
|---|---|---|
| 18:03 | 18:06 | 18:13 – 18:21 – 18:29 |
| 19:03 | 19:07 | 19:09 – 19:32 – 19:40 |
| 23:00 | 23:05 | 23:14 – 23:30 – 23:46 |

comparison. The radiosonde profiles have a fine vertical resolution. Therefore, to compare with FIRMOS L2 products, their
readings were convolved with the FIRMOS Averaging Kernels (Rodgers, 2004, AK).

Each radio sounding acquired on the $5^{th}$ of February was compared to the average of the three FIRMOS water vapour and temperature retrieved profiles nearest in time (Figure 19 and 20). Each plot refers to a different radiosonde acquisition, the local time is also reported. The retrieved products are the red curves and radiosonde profiles before and after the convolution with the FIRMOS AK are the orange and green curves, respectively, a-priori profiles are in grey. FIRMOS and a-priori retrieval
errors are also reported.

Figure 19 shows how the water vapour profiles retrieved from FIRMOS observations agree with the convolved radiosonde profiles within the retrieval errors. The third radiosound at the surface is an exception that is probably related to the different





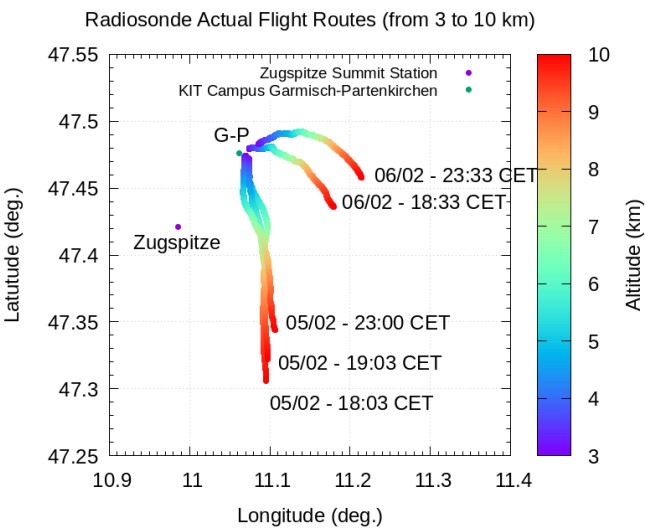

**Figure 18.** Radiosonde actual flight routes limited between 3 Km to 10 Km. The launch times of the balloons in local time are also reported.

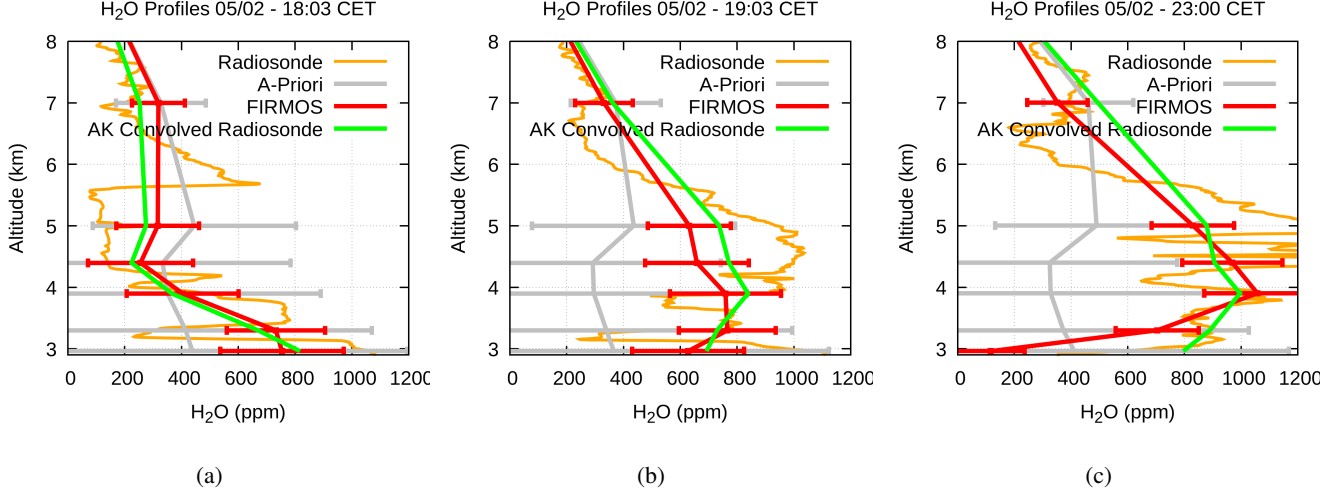

**Figure 19.** Comparison between FIRMOS L2 water vapour product (red curves, mixing ratio) and radiosonde profiles (orange curves: raw data, the green curves are convolved with the FIRMOS AK); a-priori profiles are coloured grey. FIRMOS and a-priori retrieval errors are also reported. Each plot refers to a different radiosonde acquisition, the local time of the launch is also reported.



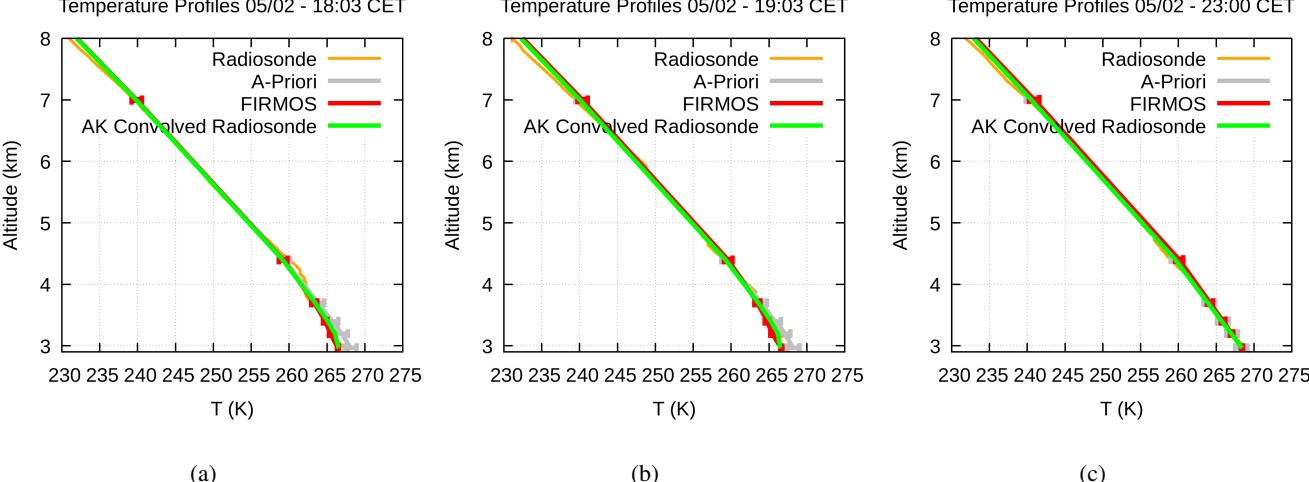

**Figure 20.** Comparison between the FIRMOS L2 temperature product (red curves) and radiosonde profiles (orange curves: raw data, the green curves are convolved with the FIRMOS AK); a-priori profiles are coloured grey. FIRMOS and a-priori retrieval errors are also reported. Each plot refers to a different radiosonde acquisition, the local time of the radiosonde launch is also reported.

boundary conditions experienced by the radiosonde during its trajectory relative to those measured by FIRMOS above the Zugspitze site. Similarly to water vapour, the comparison between the temperature obtained from FIRMOS and the convolved
radiosonde profiles shows good agreement within the FIRMOS retrieval errors.

### 4.2.2 Comparisons with Raman lidar measurements at UFS

On 5 and 6 February 2019, a total of four water vapour measurements were carried out with a high-power Raman lidar at UFS. In this period, the stratospheric aerosol lidar was continuously recording backscatter profiles in order to detect the presence of thin cirrus clouds . The lidar systems are described in detail by Klanner et al. (2021, see also Trickl et al. 2020b for more
technical details). For water vapour the system features a range from 3 to 20 km a.m.s.l. for a measurement time of one hour. The data evaluation procedure was recently refined, yielding a better agreement than described by Klanner et al. (2021) with the reference measurements of the campaign. A range extension of up to 25 km could be achieved for measurements with minimal background noise.

The water vapour mixing ratios retrieved from the lidar were calibrated by balloon-borne cryogenic sensors (CFH) of the
Forschungszentrum Jülich. The agreement of the lidar measurement with the CFH data was outstanding below 5 km, in the upper troposphere and lower stratosphere in the case of the best time overlap. Between 5 and 8 km the water vapour mixing ratio exhibited an increasingly spiky humidity structure that was different for lidar and sonde. This is explained by several spatially confined and highly variable dry layers of stratospheric air, unprecedented in spatial inhomogeneity in our lidar sounding over several decades (e.g., Trickl et al., 2014, 2020a, b, and references therein) making the instrument comparison particularly
difficult (see Vogelmann et al., 2011, 2015).




The best agreement can be expected for the vertical measurement on the summit and at UFS since the observation volumes almost match. For the lidar, we assume an uncertainty of the order of 5% on the first two days, given the excellent specifications for the CFH sondes. On the third day the uncertainty can be higher because of the rather distant calibration source.

The lidar acquisitions were compared to water vapour profiles retrieved from FIRMOS coincident measurements. The comparison was performed averaging the profiles from 5 FIRMOS observations for each Raman profile. Given the finer vertical resolution of the Raman profiles they were convolved with the AK to compare them with FIRMOS L2 products.

Figure 21 shows the comparison between the profiles from FIRMOS L2 water vapour and Raman profiles. Each plot refers to one of the FIRMOS–Lidar pairs. The retrieved products are plotted in red, the original Raman profiles in orange, and the green curve is the result of the convolution of the Raman profile with the FIRMOS AK. A-priori profiles are shown in grey. FIRMOS, Raman and a-priori retrieval errors are also reported. From Figure 21 we can conclude that the water vapour profiles retrieved from the FIRMOS observation agreed with the convolved Raman profiles within the retrieval error.

### 4.2.3  E-AERI products comparison

The KIT algorithm for the retrieval of Integrated Water Vapour (IWV) was applied to both the FIRMOS and E-AERI datasets for comparison. IWV is retrieved by minimising E-AERI (or FIRMOS) versus the Line-By-Line Radiative Transfer Model (LBLRTM, Clough et al., 2005) spectral residuals in the range from 400 cm$^{-1}$ to 600 cm$^{-1}$ (see Sussmann et al., 2016, for details). The dominant contribution to IWV precision error is the retrieval noise: the higher uncertainty value for FIRMOS precision (0.027 mm) compared to E-AERI (0.020 mm) is related to the higher NESR of FIRMOS compared to E-AERI: $\sim 2$ mW/(m$^2$ sr cm$^{-1}$) and $\sim 0.5$ mW/(m$^2$ sr cm$^{-1}$), respectively.

Note that the lower NESR in E-AERI spectra may be explained by E-AERI using a cooled detector (67 K), while FIRMOS uses a room-temperature detector. H$_2$O continuum and line parameters used in the forward calculation, as well as a-priori assumptions on the shape of the H$_2$O profile (the NCEP reanalysis as for the FIRMOS L2 data) are factors impacting the accuracy of the IWV retrieval; however, they are common to E-AERI and FIRMOS retrievals and can therefore be disregarded for the IWV intercomparison. Other factors are specific to the instruments and can cause biases between E-AERI and FIRMOS:

– altitude difference of 4 m between the E-AERI and FIRMOS location;

– frequency shifts in either or both E-AERI or FIRMOS spectra;

– calibration errors.

The impact of the altitude difference on IWV (E-AERI 2961 m asl, FIRMOS 2957 m a.m.s.l.) was corrected by calculating IWV at the two altitudes from the NCEP profile used as retrieval a-priori. The resulting difference used for the altitude correction is 0.002 mm for the mean atmospheric state of the campaign, and therefore, the error introduced by this altitude correction should be $\ll 0.002$ mm.

In addition, FTS measurements can show small errors in the frequency scale due to tiny drifts of the calibration laser. As the measured spectrum is fitted to a theoretical spectrum, such frequency errors can propagate to IWV errors in the retrieval





(a)

(b)

(c)

(d)

**Figure 21.** Comparison between the FIRMOS L2 water vapour product (red curves) and the raman profiles (with green curves and without orange curves the convolution with the FIRMOS AK). FIRMOS, a-priori and Raman retrieval errors are also reported. Each plot refers to a different Lidar acquisition and the CET time of the acquisition is also reported.




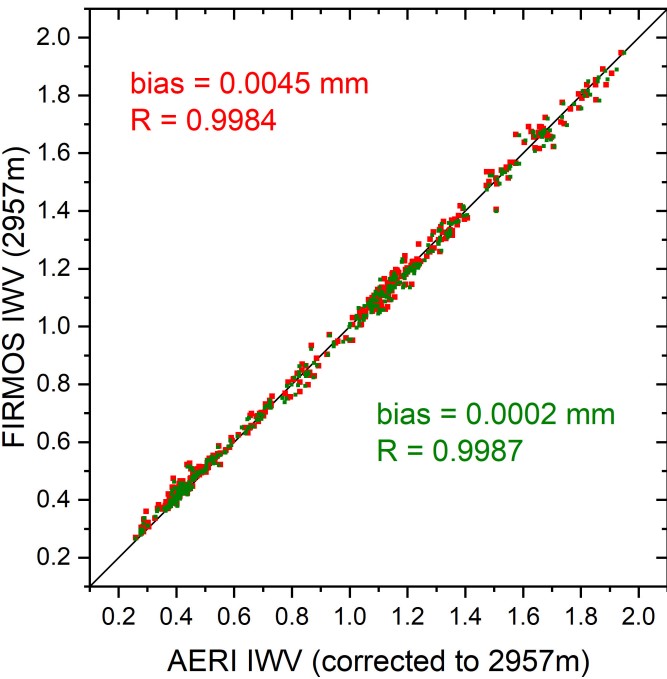

**Figure 22.** IWV values calculated for the two instruments. Red: IWV retrieval from AERI spectra and FIRMOS spectra with frequency scale as is. Green: IWV retrieval from E-AERI and FIRMOS including a joint fit of frequency scale factors (for both instruments independently).

process. Infact, direct comparison of coincident FIRMOS and E-AERI spectra ($\Delta t \leq 4$ min) showed evidence of a small discrepancy in frequency scales. Therefore, we implemented a joint fit of a frequency scale factor = 1 + frequency shift within

395   our IWV retrieval. The resulting mean wavenumber scale factor is 1.0000555 for FIRMOS and 0.9999513 for E-AERI.

The impact from this joint frequency scale retrieval on IWV is shown in Figure 22. The IWV retrievals with the original spectra are displayed in red and there is a bias of $\delta$IWV(FIRMOS-AERI) = 0.0045 mm. For IWV retrievals with joint frequency scale fit (green) the bias is practically eliminated to $\delta$IWV(FIRMOS-AERI) = 0.0002 mm. This bias of 0.0002 mm is negligible compared to the level of measured atmospheric IWV states (from 0.2 to 2 mmH$_2$O); i.e., there are no indications of significant

400   calibration errors in the spectral domain of the H$_2$O rotational band.

### 4.2.4   ERA5 reanalysis comparison

During two intervals of the 2019 campaign, between 6:00 pm on 22 January and 6:00 am on 23 January, and successively between 0:00 am on 5 February and 6:00 am on 7 February, FIRMOS observations were sufficiently frequent to create a time-series. In order to gain sufficient density the L2 retrieval results from clear sky scenes were processed together with the cloudy

405   observations analysed in Di Natale et al. (2021).

L2 products can be compared to the corresponding ERA5 reanalysis data (Hersbach et al., 2018) after a suitable lowering of the time resolution of the FIRMOS products (ie, 1 hour averaging in time zones, the temporal resolution of ERA5 data)





(a)

(b)

**Figure 23.** Water vapour time series, (left) from 22 January, 6 p.m. to 23 January 6 a.m. and (right) from 5 February, midnight to 7 February, 6 a.m., 2019: (a), profiles retrieved from FIRMOS measurements, at lowered resolution, the time frequency is about 10 measurements per hour. (b): water vapour profiles for the ERA5 pixel containing the Zugspitze station.



and interpolation on the ERA5 pressure grid (Figure 23b). The FIRMOS water vapour distribution time series is shown in Fig. 23a. It can be observed that the FIRMOS series shows higher variability compared to the ERA5 reanalysis on both the altitude and the time dimension. Such variability is probably due to local conditions that cannot be represented in ERA5 due to its larger spatial resolution (0.25° in longitude and latitude, ∼34 km). However, the two have similar trends in time, suggesting an increase in humidity at lower altitudes towards the early hours of $23^{st}$ January and $5^{st}$ February.

## 5 Conclusions

In this paper, we describe the FIRMOS Fourier transform spectroradiometer, and its performance in detecting the downwelling spectral radiance emitted by the atmosphere. FIRMOS was developed at INO-CNR to support the FORUM mission, which will be launched by ESA in 2027. FIRMOS was used to validate the measurement method and preliminary instrument design concepts by providing real measurements acquired during a field campaign, the data were used to support the feasibility studies of the mission.

FIRMOS is a spectroradiometer designed with an optical layout based on a double-input and double-output Mach-Zehnder configuration and is capable of measuring the atmospheric radiance in the spectral band from 100 to 1000 $cm^{-1}$(10–100 $\mu$m wavelength) with a spectral resolution of 0.3 $cm^{-1}$. Its measurement range, in particular, covers the pure rotational band of water vapour in the FIR region, below 600 $cm^{-1}$($\simeq$16 $\mu$m), allowing to improve the retrieval performance of water vapour as well as cloud microphysics. The dominant spectral noise on the calibrated spectrum (NESR) is on average equal to 0.002 W/m$^2$-sr-cm$^{-1}$.

To sound the upper part of the atmosphere, this kind of measurement needs to be performed in extremely dry sites, for example, at high altitude. For this reason, between December 2018 and February 2019, FIRMOS was deployed on the summit of Mount Zugspitze (Germany) at 3957 m a.m.s.l. This site is equipped with several instruments that were used to validate the FIRMOS measurements. In particular, an E-AERI spectrometer is permanently installed at the site, allowing the FIRMOS measurements to be validated in the common spectral range. Furthermore, to validate atmospheric retrieved parameters, such as the temperature and water vapour profiles, a set of radiosondes were launched during the 5 and 6 February from Garmisch-Partenkirchen, 8.6 km to the north-east of the summit, while a Raman lidar was operating at the same time from the UFS at 2675 m a.m.s.l., 700 m to the south-west of the summit station.

The retrieval from the FIRMOS spectral radiances was performed with the KLIMA retrieval code. First, a specific algorithm was implemented to select the measurements in clear-sky conditions. The algorithm first performs a linear fit in six selected microwindows in the atmospheric window (820–980 $cm^{-1}$) and then minimises the average of the ratio between the signal and the total error (quadratic sum of the NESR and calibration error) within the interval (829–839 $cm^{-1}$), chosen because of the absence of gas absorption lines, simultaneously with the slope of the linear fit. These criteria were found sufficiently reliable to select the spectra in clear-sky conditions or in the presence of very thin cirrus clouds, which affect the measurement only with a signal below the noise threshold. A set of 625 spectra out of 838 were flagged as clear-sky and, hence, analysed with the KLIMA code.



We found that the average and the standard deviation of the differences between the measured spectra and the simulations, obtained from the retrieval over the entire clear-sky dataset, is comparable with the FIRMOS calibration error and NESR, respectively, meaning that the instrument NESR and calibration error estimates are well characterised. The vertical distributions of water vapour and temperature were retrieved from FIRMOS observations by using 6 and 7 atmospheric levels, respectively,
starting from the surface up to 7 km a.m.s.l. We noted that FIRMOS measurements showed a strong variability of information content, in particular water vapour showed variations from 2 to 4.5 DOFs depending on the water vapour content in the atmosphere.

The retrieval profiles were found in very good agreement both with the profiles provided by the radiosondes and the Raman lidar. The radiosondes were launched on 5 and 6 February 2019, but during the day 6 cirrus clouds passed over the site during
the measurements. Comparisons of the retrieved water vapour and temperature profiles with the radiosoundings convolved with the averaging kernels showed that all fitted parameters lie within the retrieval error bars, with the exception of the very first level of water vapour of the last measurements. The latter was caused by radiosondes being launched from a site too far away from the Zugspitze summit.

The comparisons with the Raman profiles of water vapour, gave similar results: while for the measurements starting at
18:49 and 23:39 CET on 5 February and 19:25 on 6 February the differences between the retrieved values and the convolved radiosoundings are within the error bars, FIRMOS retrieval overestimated the very first level on the day 5 February at 19:54 CET; also in this case, this discrepancy can be explained by the non-exact co-location of the instruments.

In addition, the FIRMOS measurements were validated by comparing the retrieved IWV values with those obtained from the spectra of the E-AERI spectroradiometer, placed next to FIRMOS. We found a correlation index equal to 0.9986 and a very
low bias between the retrieved IWV estimated about -0.00007 mm. This is another confirmation that the FIRMOS and E-AERI spectral measurements are equivalent in their common spectral range.

Finally, the trends of the retrieved water vapour and temperature profiles over time were found to be in good agreement with those provided by the ERA5 reanalysis over the Zugspitze for the period of the FIRMOS campaign. The advantage of the FIRMOS observations is the higher time resolution of 1 minute compared to ERA5 (1 hour), allowing to catch faster
atmospheric cycles.

In the future, it is planned to adapt FIRMOS to stratospheric balloon platforms to provide measurements very similar to those that will be delivered by FORUM. This will require to improve instrument subsystems for near-vacuum operations and to cover the full spectral range from 100 to 1600 $\mathrm{cm}^{-1}$in order to prepare a facilities for cal/val activity of the satellite.

*Data availability.* The full dataset of the 2-month campaign, including infrared spectra (FIRMOS and E-AERI) and all the additional infor-
mation (lidars, dedicated RS), is available via the ESA campaign dataset website https://earth.esa.int/eogateway/campaigns/firmos (Palchetti et al., 2020a, https://doi.org/10.5270/ESA-38034ee). ESA requires a free registration to inform users about issues concerning data quality and news on reprocessing. Information about the data formats are reported in README files within each data sub-directory.



*Author contributions.* LP designed the experiment and was chief scientist for the field campaign, RS was responsible for the local deployment. MB, GB, FDA, AM, SV and LP designed, built and characterised FIRMOS and carried out the Zugspitze campaign. CB and LP
performed the L1 analysis. GDN implemented the clear-sky selection algorithm. SDB, MG, and GDN performed the L2 analysis. CR performed the radio soundings and processed their measurements, and SDB and GDN performed the intercomparison analysis. HN and TT performed and processed the Lidar measurements, and SDB and GDN performed the intercomparison analysis. RS and MR performed and processed the E-AERI measurements and carried out the IWV intercomparison. FB performed the comparison with the ERA5 reanalysis. CB, GDN, FP, and LP prepared the manuscript, CB coordinated the contributions from all co-authors. All authors commented on the manuscript.

*Competing interests.* One author is member of the editorial board of journal Atmospheric Measurement Techniques. Authors have no other competing interests to declare.

*Acknowledgements.* This study was supported by the European Space Agency (ESA) with the FIRMOS project (ESA–ESTEC contract no. 4000123691/18/NL/LF) for the development and the deployment of FIRMOS, as well as by the Italian Space Agency (ASI) with the research project "FORUM Scienza" (grant no. 2019-20-HH.0) supporting the scientific exploitation of the future FORUM measurements.
The dedicated balloon activities were partly supported by funding from the Helmholtz Association in the framework of MOSES (Modular Observation Solutions for Earth Systems).

Hersbach et al. (2018) was downloaded from the Copernicus Climate Change Service (C3S) Climate Data Store.





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
