# Peer review of "The Far-Infrared Radiation Mobile Observation System for spectral characterisation of the atmospheric emission"

_EGUsphere, 2022_

## Author Comment (AC1)

We thank the Referee for their comments and suggestions. In the following we answer the Referee comments point by point:

**This paper would greatly benefit from having a comparison with both the E-AERI in radiance space. Due to the differences in the spectral resolutions, I would recommend the authors use the "double difference technique" outlined in Tobin et al. JGR. 2006. As the two instruments are essentially collocated (although vertically offset by 4 m), the spectral differences between 405 and 600 cm-1 should be within the instrument noise (if both systems are well calibrated).**

We thank the referee for the suggestion to use the "double difference" technique. The analysis was added to the manuscript (Sect. 4.2.3). As in Tobin et al. we looked at the differences distribution and found an average of 0.17 mW m-2 sr -1 cm and a std-dev 1.13 mW m-2 sr-1 cm. (only the numerical details were added not the following plot)

[Figure]

**Line 253: did you assume any cross-level covariance in your a-priori? Were there any cross-correlations between temperature and humidity? There should certainly be cross-level correlations in temperature due to the atmospheric lapse rate, and a long analysis of radiosonde data from the region (or ERA5 data) should indicate if there should be other correlations in the a-priori. If you assume the a-priori is a diagonal matrix, that will essentially increase the information content (DFS) of the retrievals.**

To clarify the assumptions on the a-priori covariance matrix the following sentence was added:

"The a-priori covariance matrix was constructed assuming for both parameters a correlation length equal to 2 km between adjacent levels, while no cross-correlation was imposed between temperature and humidity".

**Line 260: It is important to note that the gradient in a cloud-free measurement is zero only because it is so dry at the Zugspitze location. If you were in a tropical location, there would be a negative slope. This needs to be stated.**

We agree with the Referee, the following was added to the text:

"[...] cloud-free measurement would have a gradient of 0 in the very dry winter conditions at Zugspitze,"

**Line 279: In the selection of the 625 cases, did the Raman lidar (or the backscatter lidar, which was briefly mentioned later in the paper) confirm that these were cloud-free?**

Yes, the backscattering lidar shows that in case of spectra selected as clear (analysable by KLIMA), no clouds occurred, on the contrary they occurred in case of spectra selected as cloudy (not analysable by KLIMA). In Fig. 1 are shown two FIRMOS spectra selected by the algorithm as clear sky (left panel) and the backscattering lidar signal (right panel) confirms the absence of clouds.

In Fig. B two FIRMOS spectra flagged as cloudy are shown. Here as well, the backscattering lidar signal confirms the occurrence of clouds.

[Figure]

Fig. A. Left panel: FIRMOS spectra selected in the presence of clear sky. Right panel: backscattering lidar signal corresponding to the FIRMOS measurements time.

[Figure]

Fig. B. Left panel: FIRMOS spectra selected in the presence of clouds. Right panel: backscattering lidar signal corresponding to the FIRMOS measurements time.

**Line 288: it was not clear if the uncertainty used in the retrievals was the NESR or the sum of the NESR and the CalErr. Please clarify this in the text. If the latter, then the chi-squared term being less than 1 could be due to the very conservative estimate of the thermistor error in the blackbodies (stated on line 230).**

We, added the following in the text:

"total error (quadratic sum of the NESR and calibration error)".

In the figure 12 and 13 (now 14 and 15) we demonstrated that the mean of the residuals reproduces well the calibration error, while the SD of the residuals is smaller than the NESR, this is the reason we are convinced NESR and not the calibration error is overestimated.

Moreover, if we set the calibration error, to zero the retrieval cannot reach the chisquare=1 because the calibration error is small with respect to the NESR.

**Line 293: the mean residual also will contain any bias error in the forward model (not only instrument calibration error).**

That is right, we added in the text:

"or forward model error"

For the two comparisons in Fig 15: it would be nice to include the integrated water vapor (IWV) amount for the two cases. Also, for line 309, the authors suggest that the DOF depends on the surface water vapor content, but it is really dependent on IWV? Turner and Löhnert (JAMC, 2014) showed that the DOF in the water vapor retrieval using AERI observations in the 538-588 $cm^{-1}$ region depends on IWV.

The value of IWV was added in the figure (now Fig. 17)

We also added a reference to Turner and and Löhnert 2014.

Figure 20: please replot using a skew-T approach, so that differences of a few degrees can be more easily identified and quantified.

The following are the T profiles using skew-T

[Figure]

Referee2 suggested to plot the T profiles as difference. The following plots show T profiles as difference wrt the a-priori ($\hat{x} - x_a$).

[Figure]

We think the difference plots are more intuitive, and clearly show the FIRMOS T profile and the AK convolved Radiosonde are within the retrieval error.

**Fig 23 and resulting analysis: this is pretty unsatisfying. I realize the purpose is to show that the FIRMOS is capturing the evolution of the event well, but the very coarse resolution of the ERA5 data in a mountainous region is totally inadequate to the task. I highly recommend that the comparison be made against higher-resolution NWP output, such as the (order)2-km resolution ICON data from the DWD. And that the figure include a subpanel showing the bias and RMS difference between the NWP model and the FIRMOS.**

We contacted DWD enquiring for data for January and February 2019, unfortunately they only provide real-time NWP data. We also followed their recommendation to contact the Fraunhofer Institute IEE for historical data, unfortunately they only have data from 2021.

We removed the section 4.2.4 (comparison with ERA5). As suggested by Ref2 we added the time-series of H2O to the end of section 4.1 (retrieval of geophysical parameters) the time resolution was increased (10 minutes) we deem the plot gives an overview of the dataset in two periods of the campaign when the acquisitions were sufficiently dense and continuous.

Question: Why did the authors not perform KLIMA retrievals using the E-AERI spectra, and then compare the retrievals from the E-AERI with the FIRMOS? This seems like it would be a relatively simple comparison and include a lot more data (e.g., there seems to be hundreds of points in Fig 22), and open up an interesting discussion because their spectral differences between the two instruments.

To share the efforts between the different groups involved in the paper it was preferred to retrieve IWV with IMK algorithm.

In view of the additional analysis suggested by the referee using the Tobin 2006 technique added to Sect 4.2.3 we believe we have shown the two instruments agree very well in the H2O spectral band.

---

## Author Comment (AC2)

We thank the Referee for their comments and suggestions. In the following we answer the Referee comments point by point:

**Section 2.1.2 Radiometric Calibration Unit**

This section was revised to clarify some points and to remove some inaccuracies noticed by the Referee.

**1) This section is not clear to me. I think I can follow the description of the blackbody setup. However, a sketch showing the positions of the 4 sensors and indicating which sensor is T1, T2, and T3, respectively, would be helpful.**

A figure was added (Figure 3), indicating the position of the sensors inside the BBs.

To avoid misunderstanding we removed (from the figures, captions, and text) the symbols T1, T2 and T3 and in the paper we refer to the different temperature sensors as PT100, NTC, Dallas1 and Dallas2, as described in Figure 3.

**2) I would also be grateful for a few words on how the temperature stabilisation is realised. How are the BBs heated/cooled?**

 The following was added to explain how temperature stabilization is obtained:

"The NTC temperature is used for the thermal stabilisation of the BBs. Each stabilisation controller (which can be turned on/off) is equipped with a Proportional Integral Derivative (PID) circuitry to maintain the temperature read from the NTC, equal to a selected value. The HBB controller operates in heating-only mode by driving a heater resistor mounted inside the HBB. The CBB controller operates in cooling/heating mode by driving a Peltier element placed inside the CBB."

**3) How is the thermal homogeneity assured?**

The thermal homogeneity was obtained thanks to the assembling of the BBs, as described in the amended text:

"Monte Carlo numerical calculations were performed to optimise the cavity geometry of the BBs, in order to maximise normal emissivity, a 34 ∘ angle was chosen for both the HBB and CBB inner cones (see Fig. 3) achieving an emissivity > 0.9985.

The CBB was assembled in a 3D-printed co-polyester plastic shell and the HBB was assembled in a 3D-printed heat resistant carbon fiber reinforced Nylon plastic shell, they were both designed to minimise thermal dispersion. The BBs cavities were fabricated in aluminium internally coated using NEXTEL-Velvet-Coating 811-21. Some layers of thermal

superinsulation foils were placed inside the plastic shells, in order to minimise the thermal exchange between the aluminium structure and its plastic supports."

About the thermal homogeneity obtained, please see our answer to the following Point 7.

**4) The accuracy of 30 mK given in Table 3 is not traceable for me. You state that you use a high-accuracy (30 mK) PT100 sensor as T1, but you do not give an accuracy for the temperature reading by the FIRMOS controller. The accuracy of the Lakeshore temperature monitor is given as 0.6%, which would correspond to 1.8 K at 300 K, which is not very likely and would make this monitor useless to correct for a possible offset of 200 mK in the FIRMOS controller. Please give more details concerning the accuracy of the different sensors and their readout electronics, also for T2 and T3.**

We apology for some inaccuracies and typos about this topic.

The accuracy reported in the original Table 3 is the PT100 accuracy, i.e. only the accuracy of the sensor without the readout electronics. We used the Lakshore monitor to make a comparison between different reading systems of the same PT100 to have an estimate of the total accuracy (sensor+readout). .

In the original text there were two inaccuracies, and we really thank the Referee for noting them:

(i) the temperature equivalent accuracy of the Lakeshore Model 218 is 68 mK (https://www.lakeshore.com/docs/default-source/product-downloads/catalog/lstc_218_l.pdf?sfvrsn=4a4f54df_7) and not "0.6%"

(ii) the difference of 200 mK between the two readout systems was not "subtracted during the signal analysis" (as erroneously written in the original text) but was considered for the calculation of the total budget of the temperature error.

The text related to this topic was completely revised, and the amended text is:

"Due to the sensor high accuracy, the PT100 is employed to measure the BB temperature value used in the L1 data analysis. The PT100 temperature reading by the FIRMOS controller and by a commercial Temperature Monitor (Lakeshore, Model 218, with a temperature equivalent accuracy of 68 mK) were compared to estimate the contribution of the readout electronics to the accuracy of the BB temperature. The comparison showed a maximum positive offset of 200 mK between the two, this value was conservatively assumed as the BB temperature total accuracy."

Moreover, in Table 3 we changed the script "Stabilization Accuracy" into "PT100 Temperature Accuracy" and we inserted a line with the total temperature accuracy (PT100+readout electronics).

**5) I am also missing a comment on the emissivity of the blackbodies. How is a deviation from 1 handled?**

The emissivity of the BBs used in FIRMOS is better than 0.9985. The effect of the deviation from one in the emissivity is negligible with respect to the error (300 mK) due to the thermal gradient. The following plot shows in blue the CalErr considering both the emissivity deviation from 1 and the 300mK error on T. The other curve is for the T error alone.

[Figure]

Amended text in Section 2.1.2

"[…] a 34 ∘ angle was chosen for both the HBB and CBB inner cones (Palchetti et al., 2008) achieving an emissivity > 0.9985."

In Section 3.1

"With this temperature error, the uncertainty due to the emissivity deviation from 1 gives a negligible contribution to the calibration error."

**6) line 146f: "the temperature of the PT100 sensor was recorded after the switching of the stabilisation." What do you mean with "switching of the stabilisation"? What is switched?**

That sentence is no longer in the text. The meaning was that we started to record the temperature at the time when the stabilization controller was switched on.

**7) line 147f: "The difference between the PT100 reading and the stabilisation temperature of each BB is reported in Figure 3 (a)-(d)."**

**No, it's not. Figs. 3(a) and 3(c) show the absolute temperature over time. Please provide a difference plot, either with respect to T1 or with respect to the stabilisation temperature (as suggested in the text). Please choose the ordinate such that the thermal inhomogeneity becomes more visible. From the plot it seems that the thermal gradient in the hot BB is rather 1 K than the 0.3 K given in Table 3, but it is hard to see from this kind of figure. Temperature variations (over time) in Figs. 3(b) and 3(d) are only given for T1 and not for each sensor as stated in the figure caption.**

We agree with the Referee that Figure 3 and the part of the text where we describe these measurements are not clear.

First, we want to specify that the PT100 measurements and the Dallas measurements cannot be compared because the accuracy of Dallas sensors is lower than the PT100 sensor, and the readout electronics are different. We use PT100 readings to measure the BB temperature while we use the difference between the Dallas readings (placed at the extremities of the BB) to infer the BB thermal homogeneity. To estimate the thermal gradient, we are not concerned with the absolute values of the Dallas sensors measurements, but only with their difference, as we verified that all the Dallas of the same model (read by the same electronics) measure the same temperature within few mK.

To avoid misunderstandings, we have split the original Figure 3 in two different figures: Figure 4 and Figure 5. Figure 4 shows the stabilization performance of the BB controllers, and it shows the PT100 measurement and the difference between the set point (with appropriate ordinate). Figure 5 shows the homogeneity performance, and it shows the Dallas measurements and the difference read by the Dallas sensors (with appropriate ordinate).

Also, the text was extended to better explain the performance and the results shown in the figures.

The amended text related to this topic is:

"Due to its high accuracy, the PT100 sensor is employed to measure the value of the BBs temperature used in the L1 data analysis " …

"Two Dallas sensors, placed at the opposite extremities of the BB, are used to monitor the BB thermal homogeneity."

"In order to find the precision of the BB thermal stabilisation, the difference between the PT100 reading and the set point (the so-called temperature stabilisation error) was recorded for some hours. Figure 4 shows the PT100 measurements after stabilisation was activated, for the HBB (Fig. 4a ), and the CBB (Fig. 4c); Figure 4b and 4d show the temperature stabilisation error after the set point is reached, respectively for the HBB and the CBB. The HBB reached the temperature of $60°$ C in approximately 2 hours and the CBB reached $15°$ C in approximately 30 minutes. To infer the precision of the temperature stabilisation, assumed as the standard deviation of the stabilisation error after the set temperature is reached, we calculated the standard deviation of the signals reported in Figure 4b and 4d. The HBB controller provides a stabilisation precision of 8.3 mK and the CBB controller provides a stabilisation precision of 1.1 mK.

The BBs temperature homogeneity was estimated from the time evolution of the difference between the readings of the two Dallas thermometers placed at the extremities of the BBs. Figure 5 shows the Dallas1 and Dallas2 measurements after stabilisation was activated for the HBB (Fig. 5a) and the CBB (Fig. 5c), and the temperature difference between the two Dallas sensors after thermal stabilisation was reached, (Fig. 5b for the HBB and 5d for the CBB). After the set temperature was reached, the HBB Dallas thermal difference did not show a significant variation and the thermal gradient remained constant with a mean value of 250 mK. The Dallas thermal difference for CBB showed only a slight decrease of about 30 mK/hour and the mean value of the thermal gradient during 4 hours resulted in 300 mK. The mean value of the temperature difference between Dallas2 and Dallas1, after temperature stabilisation was reached, was assumed to be the thermal gradient of the BBs. The BB thermal homogeneity was thus conservatively considered of about 300 mK for both."

**8) line 140 and caption of Fig. 3: In line 140 you state that the sensors T2 and T3 are of the type Dallas DS18B20, in the figure caption they are named DS60B18. Please clarify.**

The sensor used is DS18B20 and the wrong name was removed from the text.

**9) line 227f: "The corresponding calibration error CalErr is spectrally correlated but independent from one measurement to another"**

**I doubt that the blackbody temperature error is independent from one measurement to another. In contrast, I would assume that the error is dominated by systematic effects constant in time (e.g., a resistance offset or a temperature gradient). Please justify, why you can assume the calibration error as independent from one measurement to another.**

**line 229f: "which is conservatively assumed to be equal to 0.3 K."**

**In line with my comments on Section 2.1.2, I am not convinced that this is a conservative assumption. Please review this number after re-assessing the accuracy of the blackbody temperatures (and emissivities).**

We agree with the Referee that the calibration error CalError is not independent from one measurement to another. Cal Err depends on the observed scene and on the uncertainty on the theoretical Planck emission, which is dominated by the uncertainty of the temperature of the BB (as the emissivity deviation from 1 is negligible). The BB temperature uncertainty depends on two contributions: the accuracy of PT100 measurements and the BB thermal homogeneity. The first contribution is 200 mK and it is due to systematic effects independent from one measurement to the other. According to the results shown in Figure 5 b and d and to the comment at Point 7, the second contribution can be considered constant during hours of measurement and can be estimated to be 300 mK. In particular, the thermal gradient of the HBB can be estimated to be 250 mK (but calculated only for 45 minutes) and the thermal gradient of the CBB can be estimated to be 300 mK (calculated for 4 hours). We have considered a thermal gradient of 300 mK for both BBs.

Assuming 300 mK as BB temperature uncertainty is surely conservative because this value is larger than the temperature accuracy of 200 mK and it overestimates the uncertainty due to the temperature inhomogeneity, as this contribution is at the worst half of the thermal gradient.

We amended the text as follows:

"The calibration error CalErr is spectrally correlated and can be calculated through the error propagation in Eq.2 assuming as independent the uncertainty on the theoretical Planck emission of each BB ($\Delta B\_H$, $\Delta B\_C$, and $\Delta B\_R$)."

And

"The uncertainty $\Delta BH$, $\Delta BC$, and $\Delta BR$ are dominated by the uncertainty of the temperature of the BB. The BB temperature error depends on two contributions: the accuracy of PT100 measurements and the BB thermal homogeneity. As the temperature accuracy is lower with respect to the thermal homogeneity, the temperature uncertainty for all BBs can be conservatively assumed to be equal to the thermal gradient of 300 mK. With this temperature error, the uncertainty due to the emissivity deviation from 1 gives a negligible contribution to the calibration error."

**lines 275 / 278: In line 275 (and Fig. 9) you give a range of (-1,1) for the slope, while in line 278, the range of the slope is [−5·10−5,5·10−5]. Is this the same slope? Please clarify.**

We refer to the same slope, its range is [-5 10-5, 5 10-5]. In the plot the values are shown normalised this simplifies the figure. We amended the text to clarify this point.

**Section 4.1**

**In this section, I find a thorough analysis of the data in terms of fit quality and vertical resolution, but I am missing the data itself, except for two exemplary profiles. Having 625 clear sky profiles in total, it should be possible to provide a meaningful 2D plot as time series (like in Fig. 23) for both water vapour and tmperature, when cutting out the times without measurements. This would give an overview over the actual measurements and would allow the reader to comprehend the statement that the variable number of DOFs in water vapour is related to the form of the vertical profiles.**

**line 284:**

Unfortunately, the data gathered during the campaign is too discontinuous and not dense enough to satisfactorily present the entire dataset. We moved the time-series to section 4.1, the figure is at a higher time-resolution (10 mins) but on the two same periods (22-23 Jan and 5-7 Feb) when measurements were regular enough. The ordinate are now in km as the previous figures.

We removed the qualitative comparison with the ERA reanalysis (see our response to Ref1). The temperature time-series is not particularly informative (see plot below) and we don't think it should be added to the paper.

[Figure]

**A short sentence explaining the meaning of "reduced chi-square" would be helpful.**

The following sentence was added:

"the χ2 divided by the difference between the number of spectral points and the number of retrieved parameters".

**Fig. 14: Also the temperature shows a certain variability in the number of DOFs, although less pronounced than for water vapour. Do you have an explanation for this variation? Is it also related to the actual profile?**

We find that the number of DOFs for T is perfectly correlated to the the inverse of NESR (see the following plot).

[Figure]

correlation T DoF vs meanNESR (560-800 cm-1)

The following was added to the text:

"The variation in the number of DOFs for the temperature profile is due to the variation of the FIRMOS NESR, indeed, a perfect correlation between the number of DOFs and the average of the inverse of the FIRMOS NESR was found".

**line 314f:"Instead, when the effect of water vapour content on temperature retrieval is less significant, both the Averaging Kernel profiles and the vertical resolution show little variation."**

**It is not clear to me what you want to express with this sentence. Maybe the words "instead" and "when" are misleading here. Do you want to say something like: The effect of water vapour content on temperature retrieval is less significant, both the Averaging Kernel profiles and the vertical resolution show little variation (Fig. 17)."? Or do you really want to say: "\*when\* the effect of water vapour content on temperature retrieval is less significant ...". Then my question would be in which cases the effect of water vapour content on temperature retrieval \*is\* significant?**

The text now reads "Instead, the effect of IWV content on temperature retrieval is less significant, both the Averaging Kernel profiles and the vertical resolution show little variation."

**Fig. 20: Difference plots would be helpful here.**

The following plots show T profiles as difference wrt the a-priori ($\hat{x} - x_a$).

[Figure]

Referee1 suggested to use skew-T plots here, we think the difference plots are more intuitive, and clearly show the FIRMOS T profile and the AK convolved Radiosonde are within the retrieval error.

**line 355:"The agreement of the lidar measurement with the CFH data was outstanding below 5 km, in the upper troposphere and lower stratosphere in the case of the best time overlap" This is confusing, because the upper troposphere and lower stratoshere is not below 5 km. Do you mean: "... was outstanding below 5 km \*and\* in the UTLS"?**

The referee interpretation is correct.

"and" was added.

**line 413: This section reads more like a summary than a conclusion.**

We changed the section name to "discussion and conclusions" as we deem it is more appropriate. The section was partially rewritten and condensed.

**line 456: I would not call this an overestimation, if it is due to spatial variability.**

this part of the text was removed.

**line 463ff: "The advantage of the FIRMOS observations is the higher time resolution of 1 minute compared to ERA5 (1 hour), allowing to catch faster atmospheric cycles." This is a strange argument. Obviously, a local measurement is something completely different to a global set of assimilated data. I would simply omit this sentence.**

we removed the sentence.

**Technical corrections**

All the corrections suggested by the Referee were applied to the manuscript.